# Non-ventilatory events substantially contribute to rise in pulmonary arterial blood flow with early or delayed cord clamping at birth in preterm lambs

Joseph J. Smolich[1,2] 

[1]*Heart Research, Murdoch Children's Research Institute, Parkville, Victoria, Australia*
[2]*Department of Paediatrics, University of Melbourne, Parkville, Victoria, Australia*

Handling Editors: Laura Bennet & Christopher Lear

The peer review history is available in the Supporting Information section of this article (https://doi.org/10.1113/JP288749#support-information-section).

**Abstract figure legend** Lung blood flow in the baseline fetal state is characteristically low, but progressively rises with reduction of lung liquid volume and delivery of the fetus from the uterus during the birth transition. Following delivery, lung blood flow rises further after early cord clamping (ECC), with this rise preserved during a brief (<1 min) non-asphyxial interval before the start of lung ventilation. However, lung blood flow after ECC decreases to a near-baseline fetal level with a longer (>1 min) asphyxial interval before onset of lung ventilation. Lung blood flow rises rapidly with ventilation after either non-asphyxial or asphyxial ECC, reaching a similar peak newborn value by 10–15 min after birth. After delivery, lung blood flow also rises further with initial ventilation and then increases rapidly with delayed cord clamping (DCC), before attaining a newborn peak value similar to the ECC groups. Overall, the non-ventilatory events of a reduction in lung liquid volume, fetal delivery and cord clamping contribute ~30% of the increase in lung blood flow between the baseline fetal and peak newborn time points.

**Abstract** A widely held view is that an increased pulmonary arterial (PA) blood flow at birth is not triggered until the onset of lung aeration, but experimental data indicate that the non-ventilatory events of a reduction in lung liquid volume, complete fetal delivery, and umbilical cord clamping can also increase fetal PA flow. However, the effect of cord clamping strategy on the contribution of these non-ventilatory events to birth-related rises in PA flow is unknown. Accordingly, PA blood flow was measured via transit-time flow probe in anaesthetized, acutely instrumented preterm fetal lambs at baseline, after a $\sim$35% reduction in lung liquid volume, following complete fetal delivery, and then after (1) delayed cord clamping (DCC) preceded by ventilation lasting $\sim$100 s ($n = 11$), or (2) early cord clamping (ECC) followed by either a non-asphyxial ($\sim$35 s, $n = 10$) or an asphyxial interval ($\sim$100 s, $P_{O_2} < 10$ mmHg, $n = 10$) before ventilation. PA flow rose stepwise after reduction of lung liquid volume ($P < 0.001$) and fetal delivery ($P < 0.001$), as well as initial ventilation ($P < 0.001$) and subsequent DCC ($P = 0.002$). PA flow also rose after ECC ($P < 0.001$), with flow maintained in the non-asphyxial group, but markedly reduced to near-baseline fetal levels by pulmonary vasoconstriction in the asphyxial group ($P = 0.009$), before rising with ventilation ($P < 0.001$). Overall, non-ventilatory events cumulatively accounted for $\sim$30% of the fetal baseline-to-peak newborn increment in PA flow. These findings suggest that (1) non-ventilatory events substantially contribute to a perinatal rise in PA blood flow with ECC or DCC, and (2) this contribution is negated if an asphyxial level of arterial oxygenation develops after ECC.

(Received 16 February 2025; accepted after revision 11 July 2025; first published online 31 July 2025)

**Corresponding author** Joseph J. Smolich: Heart Research, Murdoch Children's Research Institute, Flemington Road, Parkville, Victoria, Australia. Email: joe.smolich@mcri.edu.au

## Key points

- Although a widely held view is that an increased pulmonary blood flow (PBF) at birth is not triggered until onset of lung aeration, reduction of lung liquid volume, complete fetal delivery and umbilical cord clamping also increase fetal PBF.
- The contribution of these non-ventilatory events to birth-related rises in PBF is unknown, particularly with different cord clamping strategies.
- Anaesthetized preterm fetal lambs instrumented with central arterial flow probes underwent birth via delayed cord clamping (DCC) preceded by ventilation, or early cord clamping (ECC) followed by either a non-asphyxial or asphyxial interval before ventilation.
- PBF rose with reduction of lung liquid volume, fetal delivery, ECC and DCC. However, while an increased PBF after ECC was maintained with a non-asphyxial interval, it fell markedly after ECC with an asphyxial interval, before rebounding with ventilation.
- Cumulatively, non-ventilatory events accounted for $\sim$ 30% of the perinatal increase in PBF occurring with DCC or ECC birth strategies.

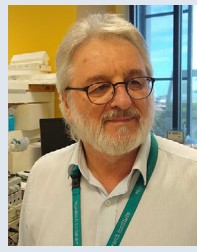

**Joe Smolich** is a clinician turned scientist who is a Principal Research Fellow within the Heart Research Group of the Murdoch Children's Research Institute in Melbourne, Australia. His research focus in recent years has been the application of methodology such as phasic blood flow, wave intensity and ventricular pressure-volume analyses to the assessment of cardiovascular function in the fetal and perinatal periods, with an emphasis on the cardiovascular effects of early and delayed cord clamping during the birth transition.

## Introduction

Pulmonary perfusion is characteristically low *in utero* under resting conditions, with blood flow measurements in fetal lambs using microspheres (Rudolph, 1985; Smolich et al., 1992; Teitel et al., 1990) or periarterial flow probes (Crossley, Allison et al., 2009; Smolich & Mynard, 2021; Smolich et al., 2011) indicating that <20% of right ventricular (RV) output flows through the lungs, with most RV output shunted right-to-left across the ductus arteriosus and into the descending thoracic aorta (Fineman et al., 1995; Rudolph, 1985). After birth, pulmonary blood flow rises rapidly to be >10-fold that of the fetus (Fineman et al., 1995; Heymann, 1999). A widely promoted current view is that this increase in flow is not triggered until the start of lung aeration (Hooper, Polglase, & Roehr, 2015; Hooper, Polglase, & te Pas, 2015; Hooper, Te Pas et al., 2015), with this paradigm primarily founded on synchrontron-based studies in rabbit kittens where a low baseline level of pulmonary blood flow measured 2–3 min after ligation of the umbilical cord was followed by a surge in pulmonary flow with commencement of ventilation (Lang et al., 2016; Lang et al., 2017).

As fetal arterial oxygenation decreases to an asphyxial level by ∼1 min after umbilical cord clamping in the absence of ventilation (Smolich et al., 2015, 2020), however, an alternate explanation for the findings of Lang et al (2016, 2017) is that baseline pulmonary blood flow was markedly reduced by low arterial oxygenation (Jensen et al., 1999; Peeters et al., 1979), and then rose rapidly following relief of this state by ventilation (Smolich et al., 2015, 2020). Thus, the paradigm that a rise in pulmonary perfusion at birth does not occur until onset of lung aeration (Hooper, Polglase, & Roehr, 2015; Hooper, Polglase, & te Pas, 2015; Hooper, Te Pas et al., 2015) may not apply to clinically relevant non-asphyxial birth transition scenarios such as delayed cord clamping (DCC) preceded by a period of ventilation, or early cord clamping (ECC) followed by a brief (<1 min) interval before the onset of ventilation (Smolich & Kenna, 2022; Smolich et al., 2015, 2017; Smolich, Kenna, Mynard, Phillips et al., 2019).

In line with this proposition, three non-ventilatory events of the birth transition are likely to contribute to perinatal rises in pulmonary perfusion. The first is a reduction in the volume of lung liquid filling the future airways, which is a normal accompaniment of the birth process (Berger et al., 1998; Pfister et al., 2001; Stockx et al., 2007). This reduction increases pulmonary blood flow via a combination of (1) diminished external compression of pulmonary microvessels (Fineman et al., 1995; Heymann, 1999; Walker et al., 1988), and (2) an elevation in RV output secondary to decreased external constraint of the heart by the fluid-filled lungs (Smolich & Mynard, 2019; Smolich, Cheung et al., 2021). The second event is complete delivery of the fetus from the uterus, which augments pulmonary arterial (PA) blood flow and RV output, while decreasing mean left atrial blood pressure (Smolich & Kenna, 2022). The third event is clamping of the umbilical cord. Thus, consistent with the observation that cord occlusion in fetal sheep transiently increased PA blood flow before this flow fell after a decline in arterial oxygenation (Campbell et al., 1967), ECC prior to ventilation at birth is followed by an initial rise in PA blood flow that precedes any subsequent asphyxia-related reduction in PA flow (Smolich & Kenna, 2022; Smolich et al., 2015, 2020). Moreover, PA blood flow also rises with DCC after a period of initial ventilation (Bhatt et al., 2013; Smolich & Kenna, 2022, 2025; Smolich et al., 2024). However, little is known about the cumulative contribution of these non-ventilatory events to rises in PA flow during the birth transition, particularly with different cord clamping strategies.

The primary aim of this study in acutely instrumented preterm lambs was therefore to quantify the contribution of a reduction in lung liquid volume, fetal delivery and cord clamping, as well as ventilation, to perinatal rises in PA blood flow in three different birth scenarios. These comprised (1) DCC preceded by ventilation, (2) ECC with a brief non-asphyxial interval prior to onset of ventilation, and (3) ECC with an extended pre-ventilation interval where arterial oxygenation fell to an asphyxial level (Smolich et al., 2015, 2017, 2020). Given that fetal and newborn rises in PA blood flow can be related to a diverse range of cardiovascular mechanisms that include an increase in RV output, a greater distribution of RV outflow to the lungs, decreased backflow of PA systolic inflow, and phasic left-to-right ductal shunting (Crossley, Allison et al., 2009; Smolich, 2014; Smolich & Kenna, 2022; Smolich & Mynard, 2019; Smolich, Kenna, & Mynard, 2019; Smolich et al., 2012b; Smolich et al., 2020), a secondary aim was to determine whether the contribution of these mechanisms exhibited a defined temporal pattern during perinatal rises in PA blood flow.

## Methods

### Ethical approval

Studies conformed to guidelines of the National Health and Medical Council of Australia (National Health & Medical Research Council, 2013), were approved by the Murdoch Children's Research Institute (MCRI) Animal Ethics Committee (Project numbers A765, A816, and A872) and were in accord with the ARRIVE guidelines for reporting of animal research (Grundy, 2015; Percie du Sert et al., 2020).

## Surgical preparation

The general features of surgical preparation were as previously described (Smolich et al., 2016, 2020). Briefly, 31 Border-Leicester cross ewes sourced from an MCRI-approved farm were acclimatized for at least 4 days in individual pens within the MCRI Translational Research Unit, with *ad libitum* access to feed and water. After fasting overnight with *ad libitum* access to water, ewes were premedicated at a gestation of 127(2) days (mean(SD), term = 147 days) using I.M. ketamine (5 mg kg$^{-1}$) and xylazine (0.1 mg kg$^{-1}$), with anaesthesia then induced via 4% isoflurane given by mask. After intubation of the trachea with a cuffed endotracheal tube, anaesthesia was maintained with isoflurane (0.5%–2%) and nitrous oxide (10%–20%) delivered in $O_2$-enriched air via positive-pressure ventilation, supplemented by an I.V. infusion of ketamine (1–1.5 mg kg$^{-1}$ h$^{-1}$), midazolam (0.1–0.15 mg kg$^{-1}$ h$^{-1}$) and fentanyl (2–2.5 μg kg$^{-1}$ h$^{-1}$). Transcutaneous oxygen saturation ($S_{pO_2}$) of ewes was monitored continuously with a pulse-oximetry sensor applied to an ear or cheek. The right common carotid artery was cannulated for monitoring of blood pressure and for blood gas analysis (ABL800, Radiometer, Copenhagen, Denmark), with ventilation adjusted to maintain arterial $P_{O_2}$ ($P_{aO_2}$) at 100–120 mmHg and arterial $P_{CO_2}$ ($P_{aCO_2}$) at 35–40 mmHg.

The uterus was exposed via a midline laparotomy. With a multiple pregnancy, the position of all fetuses was assessed by palpation, with the presence and degree of any meconium staining determined in each fetus via a small keyhole incision, usually over a hindlimb in a uterine horn. The most accessible fetus in the best condition was then chosen for surgical preparation, with any other fetus(es) first completely delivered from the uterus and humanely killed with an intracardiac injection of sodium pentobarbitone (100 mg kg$^{-1}$) after the umbilical cord had been clamped and cut.

The head of the fetus undergoing surgical preparation was placed in a saline-filled surgical glove to prevent loss of lung liquid. After exteriorization of the left forelimb and adjacent thorax, fluid-filled catheters were passed into the superior vena cava via the left axillary ($n = 16$) or external jugular vein ($n = 15$) for fluid and drug administration. Through either the left axillary ($n = 17$) or common carotid artery ($n = 14$), a fluid-filled catheter was introduced into the brachiocephalic trunk for pressure measurement and blood sampling, and a 3.5-Fr micromanometer catheter (SPR-524, Millar Instruments, Houston, TX, USA) was passed into the ascending aorta/aortic trunk to obtain a high-fidelity pressure signal. Subsequently, a left thoracotomy was performed in the 3rd interspace, with major central arteries carefully dissected for placement of non-constrictive transit-time flow probes (Transonic Systems, Ithaca, NY, USA) around the ductus arteriosus (8 or 10 mm) and left pulmonary artery (4 or 6 mm). To measure pressures, a fluid-filled and another 3.5-Fr micromanometer catheter were inserted via purse-string sutures into the pulmonary trunk close to its junction with the ductus and common pulmonary artery, and a fluid-filled catheter was introduced into the left atrial appendage. Finally, a clamped 4.5 mm endotracheal tube filled with normal saline and containing a side-port for measuring pressure was inserted via a tracheostomy, and the glove over the fetal head then removed. Due to the presence of multiple catheters and flow probes, chest wall and intrathoracic structures were positioned in their normal anatomical location at the end of surgery, but the thoracotomy was not surgically closed. Haemodynamics were allowed to stabilize for ~10 min after completion of surgery.

## Experimental protocol

After withdrawal of a 0.4 ml aortic trunk sample from the fetus for blood gas analysis (ABL800, Radiometer), a ~20 s period of steady-state baseline physiological data was collected onto a computer. To simulate a reduction in lung liquid volume that occurs during the birth process (Berger et al., 1998; Pfister et al., 2001; Stockx et al., 2007), the endotracheal tube was then unclamped and lung liquid allowed to drain passively via gravity over ~30 s into a measuring cylinder until drainage largely stopped. After the endotracheal tube was re-clamped, taking care that the lumen remained entirely filled with fluid and that no air bubbles entered the lungs, an aortic sample was again withdrawn for blood gas analysis and another ~20 s block of data acquired. While signals were recorded continuously, the fetus was then completely delivered from the uterus, placed on the ewe's abdomen and covered with warmed towels, avoiding tension on the umbilical cord or a change in the position of the fetal head and upper thorax. An aortic sample was withdrawn ~30 s after delivery for blood gas analysis.

After delivery, lambs underwent one of three birth transition protocols allocated before the start of surgery. The first was DCC preceded by ventilation for 103(13) s ($n = 11$; 6 males and 5 females; 2 singletons, 8 twins and 1 triplet), where the endotracheal tube was unclamped and connected to an infant ventilator (SLE5000, SLE Ltd, Croydon, UK), with aortic samples withdrawn for blood gas analysis 30 s after the start of ventilation, and just prior to cord clamping. The second was ECC with a brief, non-asphyxial cord clamp-to-ventilation interval ($n = 10$; 5 males and 5 females; 1 singleton, 8 twins and 1 triplet), where cord clamping was followed by ventilation 36(8) s later, with sampling of aortic blood at 20 s after ECC. The third was ECC with an asphyxial cord clamp-to-ventilation interval ($n = 10$; 5 males and

5 females; 9 twins and 1 triplet), where ventilation was started 100(14) s after cord clamping, with aortic blood samples withdrawn 30 s after ECC, and just before the start of ventilation. In all groups, aortic samples were also collected for blood gas analysis at 0.5, 10 and 15 min after birth, which was defined as the point where both ventilation and cord clamping had occurred.

All groups underwent volume-targeted positive-pressure ventilation using warmed and humidified gases, with initial settings comprising an end-expiratory pressure of 8 $cmH_2O$, a maximal peak inspiratory pressure of 50 $cmH_2O$, a respiratory rate of 60 breaths $min^{-1}$, an inspiratory time of 0.4 s, a tidal volume of 7 ml $kg^{-1}$ estimated body weight and a fractional inspired $O_2$ concentration of 0.3. These settings are physiologically appropriate for the gestational age of lambs used in the present study (Tingay et al., 2014). Settings were unchanged within the initial minute after birth, with ventilation then adjusted to increase pre-ductal $S_{pO_2}$, measured with a pulse oximetry sensor on the lamb's cheek or left forelimb, to 85%–95% by the 10 min time point (Wyckoff et al., 2015). After birth, anaesthesia in newborn lambs was continued via an i.v. infusion of ketamine (4–8 mg $kg^{-1}$ $h^{-1}$) and midazolam (0.05–0.1 mg $kg^{-1}$ $h^{-1}$).

Animals were humanely killed with i.v. pentobarbitone (100 mg $kg^{-1}$), which was administered to ewes after birth of lambs, and to lambs after completion of the study protocol. At postmortem, lambs of the three study groups did not differ in body weight ($P = 0.968$, one-way analysis of variance, average 3.58(0.52) kg), total lung weight ($P = 0.501$, average = 123(22) g), total-to-left lung weight ratio ($P = 0.239$, average = 2.48(0.12)), or total lung-to-body weight ratio ($P = 0.402$, average = 34.5(5.0) g $kg^{-1}$).

## Physiological data

Aortic trunk, pulmonary trunk, left atrial and tracheal catheter pressures were measured with calibrated transducers referenced to atmospheric pressure at the level of the left atrium. Signals from fluid-filled and micro-manometer catheters, as well as flow probes, were digitized at a sampling rate of 1 kHz and displayed using programmable acquisition and analysis software (Spike2, Cambridge Electronic Design, Cambridge, UK).

Fetal recordings were analysed at baseline and after reduction of lung liquid volume. Birth transition data were analysed at multiple time points which comprised (1) a ~20-s epoch after complete fetal delivery, just before ventilation in the DCC group, or cord clamping in the ECC groups, (2) a ~10-s epoch within 30 s after the start of ventilation in the DCC group, or cord clamping in the ECC groups, (3) a ~10-s epoch just prior to cord clamping

in the DCC group, or start of ventilation in the ECC groups, (4) a ~20-s epoch within the first 30 s after birth, a period when PA blood flow is relatively stable (Smolich & Kenna, 2022; Smolich et al., 2015, 2016, 2020) and yet to be fully influenced by potent vasodilator mechanisms stimulated with birth (Black et al., 1997; Cornfield et al., 1992) and (5) ~20-s epochs at 10 and 15 min after birth, a time-window when PA blood flow reaches an early neonatal peak (Crossley, Allison et al., 2009; Smolich & Kenna, 2022; Smolich et al., 2015, 2016, 2020; Sobotka et al., 2011), with values averaged.

During data analysis, 50 Hz mains electrical interference in digitized signals was removed using a 48 Hz low-pass filter. Mean aortic and pulmonary trunk micromanometer pressures were calibrated by matching to corresponding mean fluid-filled catheter pressures. RV output (i.e. pulmonary trunk blood flow) was derived as the sum of the instantaneous ductal and total (i.e. left plus right) PA flows over the entire cardiac cycle, with total PA flow computed as the product of measured left PA flow and the post-mortem total-to-left lung weight ratio (Smolich et al., 2020; Smolich, Kenna et al., 2021). Measurements were performed on ensemble-averaged signals typically generated from >20 beats in ~10-s epochs and >40 beats in ~20-s epochs. Pulmonary vascular conductance, the reciprocal of pulmonary vascular resistance, was computed as (total mean PA flow)/(mean pulmonary trunk pressure – mean left atrial pressure). Within individual animals, the percentage contribution of non-ventilatory events to the increase in PA flow during the birth transition was calculated as their cumulative rise divided by the overall increment in PA flow occurring between the baseline fetal and peak neonatal values (subtraction of this percentage from 100 yielded the contribution of ventilation to the increase in PA flow within each individual animal).

## Phasic blood flow analyses

RV systolic outflow was measured from the start of the pulmonary trunk systolic upstroke to the point where this flow returned to zero or a nadir close to pulmonary valve closure, with the latter indicated by the incisura of the high-fidelity pulmonary trunk pressure signal. PA systolic inflow was measured from the start of the PA flow upstroke to the point where this flow returned to zero in mid-to-late systole before birth, or a nadir close to pulmonary valve closure after birth. Note that (1) after birth, the portion of PA inflow derived directly from RV systolic outflow was calculated as total PA systolic flow minus any systolic left-to-right ductal shunt flow; (2) the proportion of PA systolic inflow derived from RV systolic outflow was computed as the PA-to-pulmonary trunk systolic flow ratio; (3) PA backflow was indicated

**Table 1. Aortic trunk blood gas variables at baseline, and after reduction of lung liquid volume and complete delivery of the fetus**

| Variable | Baseline | ↓Lung liquid | Post-delivery | $P_1$ | $P_2$ | $P_3$ |
|---|---|---|---|---|---|---|
| pH | 7.32 (0.02) | 7.31 (0.02) | 7.30 (0.03) | 0.030 | <0.001 | 0.832 |
| $S_{aO_2}$ (%) | 64.5 (7.7) | 62.6 (7.6) | 57.3 (6.4) | 0.096 | <0.001 | 0.795 |
| $P_{aO_2}$ 2 (mmHg) | 23.9 (2.3) | 23.5 (2.3) | 21.7 (2.1) | 0.215 | <0.001 | 0.250 |
| $P_{aCO_2}$ (mmHg) | 46.9 (2.7) | 47.3 (3.2) | 48.7 (2.8) | 0.284 | 0.003 | 0.251 |

Data are expressed as means(SD); $n = 31$. Abbreviations: $S_{aO_2}$, haemoglobin oxygen saturation; $P_{aO_2}$, arterial $O_2$ tension; $P_{aCO_2}$2, arterial $CO_2$ tension. $P_1$, baseline *vs.* lung liquid reduction values; $P_2$, lung liquid reduction *vs.* post-delivery values, repeated measures one-way ANOVA followed by contrasts with Bonferroni correction. $P_3$, treatment group effect, repeated measures two-way ANOVA.

by negative PA flow during diastole; (4) PA backflow included the prominent transient negative peak in the PA flow waveform occurring just before pulmonary valve closure (Smolich et al., 2011; Smolich et al., 2012a); (5) the relative degree of PA backflow was estimated from the PA backflow-to-inflow ratio (Smolich, 2014) and (6) all measured raw flows were multiplied by the quotient of flow duration and heart period to yield reported flow values (Smolich, Kenna, & Mynard, 2019).

To obtain the phasic left-to-right component of ductal shunting, raw measured flows of all individual negative segments in the ensemble-averaged ductal flow profile were multiplied by the quotient of segment duration and heart period to yield flow values that summed to total left-to-right ductal flow (Smolich & Kenna, 2025; Smolich et al., 2016, 2020).

## Statistical analysis

Results were analysed using GraphPad Prism 9 (GraphPad Software Inc., La Jolla, CA, USA), with logarithmic transformation of data having a non-normal distribution. As the experimental protocol up to and including complete delivery of the fetus from the uterus was identical in all groups, combined and individual group data from the baseline, post-lung liquid removal and post-delivery time points were analysed with one-way repeated measures analysis of variance (ANOVA), while group data were compared with two-way repeated measures ANOVA. Combined and individual group post-delivery and initial post-cord clamping data in the ECC groups, as well as longitudinal data after delivery and birth in all groups, were analysed using one-way repeated measures ANOVA. All one-way repeated measures ANOVA analyses were followed by partitioning of the sum of squares into individual degrees of freedom, with application of a Bonferroni correction for multiple comparisons. In the baseline, post-lung liquid reduction and post-delivery states, the relationship between mean tracheal and left atrial pressures, as well as between pulmonary vascular conductance and the PA backflow-to-inflow ratio, was analysed with least-squares linear regression. Data are

expressed as means(SD) and significance was taken at $P < 0.05$.

## Results

### Aortic blood gas variables

With reduction of lung liquid volume by 14(4) ml (kg body weight)$^{-1}$, which represented ∼35% of lung liquid volume at 128 days' gestation (Harding & Hooper, 1996), only aortic trunk pH declined ($P = 0.030$), but after fetal delivery, pH, $S_{aO_2}$, and $P_{aO_2}$ fell while $P_{aCO_2}$ rose ($P \leq 0.003$), with no difference between groups ($P \geq 0.250$, Table 1). Falls in oxygenation and a rise in $P_{aCO_2}$ progressed further after ECC and were most pronounced in the asphyxial group, but rapidly reversed with ventilation. By contrast, initial ventilation and DCC increased $S_{aO_2}$ and $P_{O_2}$ and reduced $P_{aCO_2}$ (Table 2).

### Arterial blood pressures and heart rate

Relatively minor changes in mean aortic and pulmonary trunk blood pressures (Fig. 1*A* and *B*) and heart rate (Fig. 1*C*) occurred with reduction of lung liquid volume and fetal delivery. Aortic and pulmonary trunk pressures increased after cord clamping ($P < 0.001$) and decreased with ventilation in the non-asphyxial ECC and DCC groups ($P = 0.002$), with minor alterations in heart rate. However, these variables fell in the asphyxial ECC group ($P \leq 0.027$), and then increased with the onset of ventilation ($P \leq 0.001$).

### Left atrial blood pressure

Mean left atrial blood pressure fell stepwise with reduction of lung liquid volume ($P < 0.001$) and fetal delivery ($P < 0.001$), and initially changed little after cord clamping or ventilation, but rose markedly by 10–15 min after birth in all groups (Fig. 2*A*). Furthermore, falls in left atrial pressure between the baseline and post-delivery time points displayed a linear relationship with associated falls in tracheal pressure ($R^2 = 0.52$, $P < 0.001$, Fig. 2*B*).

**Table 2. Aortic trunk blood gas variables at birth following early cord clamping with either a non-asphyxial ($ECC_{NA}$) or asphyxial ($ECC_A$) interval prior to ventilation, or after initial ventilation with delayed cord clamping (DCC).**

| Variable | Group | After delivery | $CC_1/V_1$ | $CC_2/V_2$ | $NB_1$ | $NB_2$ | $P_1$ | $P_2$ | $P_3$ | $P_4$ |
|---|---|---|---|---|---|---|---|---|---|---|
| pH | $ECC_{NA}$ | 7.30 (0.02) | 7.28 (0.02) | | 7.28 (0.02) | 7.34 (0.05) | 0.125 | | 0.944 | <0.001 |
| | $ECC_A$ | 7.29 (0.02) | 7.26 (0.02) | 7.24 (0.03) | 7.22 (0.03) | 7.28 (0.06) | 0.093 | 0.112 | 0.168 | <0.001 |
| | DCC | 7.29 (0.04) | 7.31 (0.03) | 7.33 (0.03) | 7.33 (0.04) | 7.33 (0.07) | 0.136 | 0.261 | 0.821 | 0.939 |
| $S_{aO_2}$ (%) | $ECC_{NA}$ | 57.3 (6.5) | 31.8 (8.8) | | 53.8 (21.6) | 95.2 (4.4) | <0.001 | | <0.001 | <0.001 |
| | $ECC_A$ | 55.4 (6.9) | 20.9 (6.0) | 7.4 (2.4) | 60.3 (25.8) | 96.6 (2.8) | <0.001 | 0.063 | <0.001 | <0.001 |
| | DCC | 59.1 (5.8) | 77.2 (8.2) | 83.5 (8.2) | 81.6 (10.9) | 94.5 (2.1) | <0.001 | 0.040 | 0.414 | <0.001 |
| $P_{aO_2}$ (mmHg) | $ECC_{NA}$ | 21.2 (1.6) | 15.0 (2.1) | | 21.2 (6.3) | 55.1 (25.2) | 0.302 | | 0.297 | <0.001 |
| | $ECC_A$ | 21.0 (2.1) | 12.8 (2.1) | 7.7 (2.0) | 26.2 (10.8) | 61.2 (15.7) | 0.091 | 0.152 | <0.001 | <0.001 |
| | DCC | 22.9(2.0) | 30.5 (5.2) | 35.3 (7.4) | 36.2 (14.2) | 47.1 (7.0) | 0.032 | 0.085 | 0.757 | <0.001 |
| $P_{aCO_2}$ (mmHg) | $ECC_{NA}$ | 48.9 (2.3) | 53.8 (2.5) | | 49.4 (2.9) | 41.0 (6.5) | 0.048 | | 0.080 | <0.001 |
| | $ECC_A$ | 49.1 (2.6) | 55.5 (3.1) | 58.6 (4.1) | 54.8 (5.5) | 45.6 (9.0) | 0.003 | 0.164 | 0.071 | <0.001 |
| | DCC | 48.0 (3.5) | 43.2 (3.9) | 40.2 (3.8) | 40.3 (4.0) | 41.6 (7.4) | 0.010 | 0.110 | 0.975 | 0.359 |

Data are expressed as means(SD); $n = 10$ for $ECC_{NA}$, $n = 10$ for $ECC_A$ and $n = 11$ for DCC groups. Abbreviations: $CC_1$, within 30 s after early cord clamping; $CC_2$, just before ventilation after asphyxial early cord clamping; $V_1$, within 30 s after onset of ventilation in DCC group; $V_2$, just before cord clamping in DCC group; $NB_1$, within initial 30 s after birth; $NB_2$, initial neonatal peak in pulmonary arterial blood flow at 10–15 min after birth. Other abbreviations are as in Table 1. $P_1$, post-delivery *vs.* $CC_1$ or $V_1$; $P_2$, $CC_1$ *vs.* $CC_2$ or $V_1$ *vs.* $V_2$; $P_3$, $CC_1$, $CC_2$ or $V_2$ *vs.* $NB_1$; $P_4$, $NB_1$ *vs.* $NB_2$, one-way repeated measures ANOVA followed by contrasts with Bonferroni correction.

## Pulmonary blood flow and vascular conductance

Mean PA blood flow rose from 32(26) ml min$^{-1}$ at baseline to 80(56) ml min$^{-1}$ after reduction of lung liquid volume ($P < 0.001$), and then to 179(97) ml min$^{-1}$ following fetal delivery ($P < 0.001$), with these flows corresponding to 6(5)%, 14(10)% and 25(12)% of RV output respectively. After ECC, mean PA blood flow increased to 256(129) ml min$^{-1}$ ($P < 0.001$), and then remained stable in the non-asphyxial group ($P = 0.635$), but fell to 50(51) ml min$^{-1}$ in the asphyxial group ($P = 0.009$). PA blood flow then rose after onset of ventilation ($P \leq 0.006$), with the rise greater in the asphyxial group (524(216) *vs.* 181(201) ml min$^{-1}$, $P = 0.002$). In the DCC group, PA blood flow increased to 323(112) ml min$^{-1}$ with initial ventilation ($P < 0.001$), and then to 475(137) ml min$^{-1}$ with DCC ($P = 0.012$). However, peak PA flows were similar between groups (Fig. 3*A*). These patterns of change in mean PA flow were mirrored in pulmonary vascular conductance (Fig. 3*B*).

## Right ventricular output and pulmonary arterial systolic flow

RV output rose from 501(86) ml min$^{-1}$ at baseline to 567(116) ml min$^{-1}$ after reduction of lung liquid volume ($P < 0.001$), and to 680(101) ml min$^{-1}$ following fetal delivery ($P < 0.001$). After ECC, this outflow fell to 580(90) ml min$^{-1}$ ($P < 0.001$), and was then unchanged in the non-asphyxial group, but declined further to 275(75) ml min$^{-1}$ in the asphyxial group ($P < 0.001$). With onset of ventilation after ECC, RV output changed little in the non-asphyxial group, but rose by 283(99) ml min$^{-1}$ in the asphyxial group ($P < 0.001$). By contrast, RV output was unaltered with initial ventilation and fell to 465(155) ml min$^{-1}$ after DCC ($P < 0.001$, Fig. 4*A*).

PA systolic inflow rose after reduction of lung liquid volume ($P = 0.035$), and fetal delivery ($P < 0.001$). However, this inflow changed little after ECC and remained stable in the non-asphyxial group, but more than halved in the asphyxial group ($P = 0.021$), before rising with ventilation in both groups ($P \leq 0.003$). On the other hand, PA systolic inflow rose with initial ventilation ($P < 0.001$) and was at first unaffected by DCC, but then increased ($P = 0.005$), with levels similar between groups by peak PA flow (Fig. 4*B*).

The fraction of RV systolic outflow continuing as PA systolic inflow was unaltered between the baseline and post-delivery time points ($P = 0.769$, average = 36(8)%). This fraction changed little after ECC, but increased to 58(15)% with onset of ventilation ($P \leq 0.003$), and then to 95(4)% at peak PA flow ($P < 0.001$). By contrast, this fraction rose from 36(8) to 48(7)% with initial ventilation ($P < 0.001$), to 67(11)% after DCC ($P < 0.001$) and to 96(4)% at peak PA flow ($P < 0.001$, Fig. 4*C*).

## Pulmonary arterial diastolic flow

A large PA diastolic backflow at the fetal baseline ($-136(56)$ ml min$^{-1}$) was reduced to $-108(60)$ ml min$^{-1}$ after a fall in lung liquid volume ($P < 0.001$), and to $-45(61)$ ml min$^{-1}$ following fetal delivery ($P < 0.001$). With ECC, PA backflow was replaced by forward diastolic

flow of 50(85) ml min$^{-1}$ ($P < 0.001$), which remained stable in the non-asphyxial group ($P = 0.364$), but reverted to a backflow of $-57(51)$ ml min$^{-1}$ in the asphyxial group ($P = 0.007$). In both ECC groups, PA forward diastolic flow increased with the onset of ventilation ($P \leq 0.023$), and then rose further at peak PA flow ($P \leq 0.036$). On the other hand, PA forward diastolic flow appeared during initial ventilation, increased after DCC ($P < 0.001$) and rose further at peak PA flow ($P < 0.001$), with peak values similar in the three groups (Fig. 5$A$).

The fetal PA backflow-to-inflow ratio fell from 80(16)% at baseline to 58(24)% with reduction of lung liquid volume ($P < 0.001$), and to 27(24)% after fetal delivery ($P < 0.001$). This ratio was subsequently near-zero in the non-asphyxial ECC and DCC groups, but transiently increased from 11(16) to 57(38)% in the asphyxial ECC group ($P < 0.001$, Fig. 5$B$). Moreover, the PA backflow-to-inflow ratio displayed a strong negative linear relationship with pulmonary vascular conductance during the birth transition ($R^2 = 0.84$, $P < 0.001$, Fig. 5$C$).

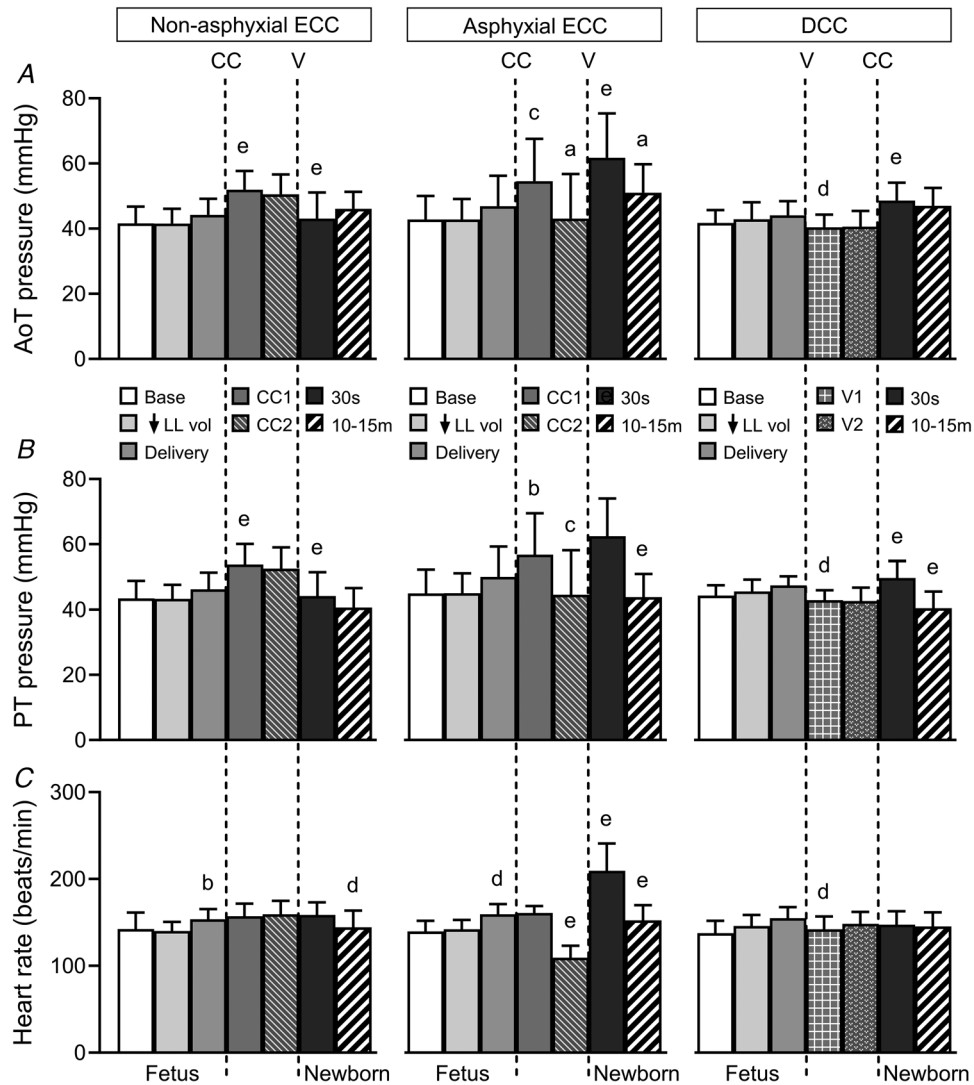

**Figure 1. Mean aortic (*A*) and pulmonary trunk (*B*) blood pressures and heart rate (*C*)**
Abbreviations: AoT, aortic trunk; PT, pulmonary trunk; DCC, delayed cord clamping; ECC, early cord clamping; CC, cord clamping; V, ventilation. Time points correspond to fetal baseline (Base), following reduction of lung liquid volume (↓LL vol), after complete delivery of fetus (Delivery), the initial 30 s (CC1) and just before ventilation (CC2) after early cord clamping, the initial 30 s after onset of ventilation (V1) and just before delayed cord clamping (V2), the initial 30 s and 10–15 min in the newborn. Data are expressed as means(SD); $n = 10$ for non-asphyxial and asphyxial ECC groups, $n = 11$ for DCC group. [a]$P \leq 0.048$, [b]$P \leq 0.019$, [c]$P \leq 0.010$, [d]$P \leq 0.004$, [e]$P \leq 0.001$, compared to preceding time point.

## Left-to-right ductal shunting

Phasic left-to-right ductal shunting was absent or trivial in the fetal baseline and after reduction of lung liquid volume, but rose to 9(16) ml min$^{-1}$ after fetal delivery ($P = 0.004$). This left-to-right shunting increased by 38(33) ml min$^{-1}$ after ECC ($P < 0.001$) and remained stable in the non-asphyxial group ($P = 0.603$), but tended to fall in the asphyxial group ($P = 0.090$)., A subsequent abrupt increase in left-to-right ductal shunting with ventilation ($P = 0.002$) was 2-fold higher in the asphyxial ECC group (122(124) *vs.* 41(36) ml min$^{-1}$, $P = 0.016$). By contrast, left-to-right ductal shunting increased to 20(21) ml min$^{-1}$ with initial ventilation ($P = 0.012$) and rose by an additional 123(67) ml min$^{-1}$ with DCC ($P < 0.001$), with the latter increment greater than after cord clamping in the ECC groups ($P < 0.001$). Nonetheless, the magnitude of left-to-right ductal shunting was similar in all three groups at peak PA flow (Fig. 6*A*). In association with these absolute flow changes, the percentage contribution of left-to-right ductal shunting to PA flow increased progressively ($P < 0.001$), with similar post-birth peaks in all three groups (average = 54(13)%, Fig. 6*B*).

## Non-ventilatory contribution to perinatal pulmonary perfusion

The cumulative non-ventilatory increments in PA blood flow accounted for 31(14)%, 28(18)% and 33(14)% of the fetal baseline-to-peak newborn increment in PA flow within the non-asphyxial ECC, asphyxial ECC and DCC groups, respectively (Fig. 7*A*). These contributions were accompanied by corresponding increments of 17(7)%, 18(10)% and 18(9)% in pulmonary vascular conductance (Fig. 7*B*).

## Discussion

This study, which evaluated non-ventilatory and ventilatory contributions to rises in pulmonary arterial (PA) blood flow during the birth transition of anaesthetized preterm lambs after early (ECC) or delayed

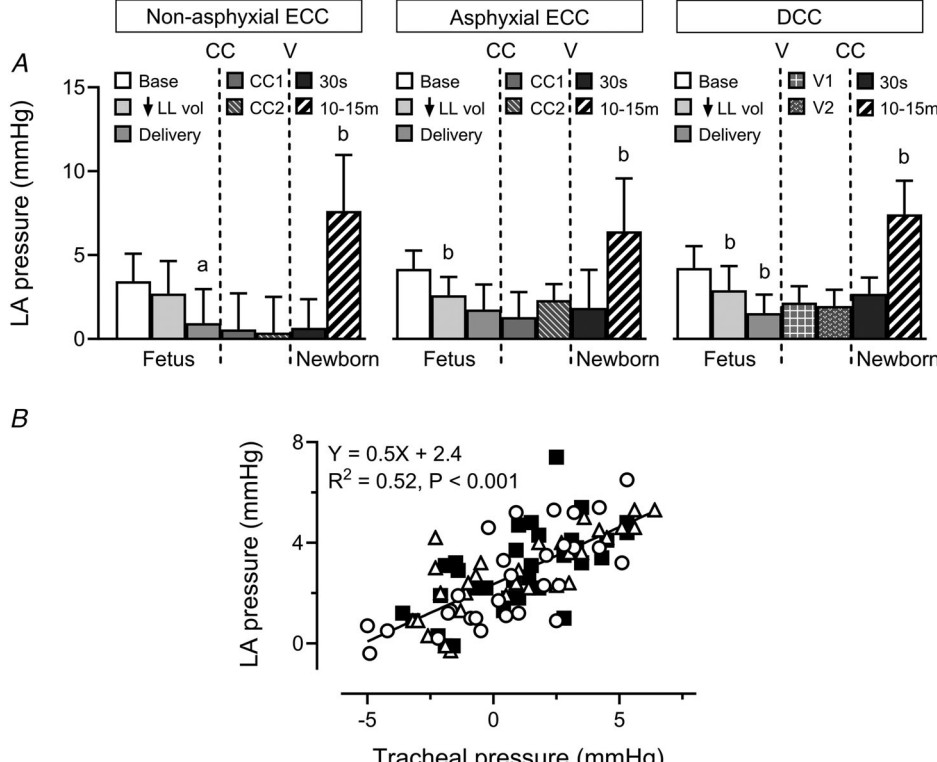

**Figure 2. Mean left atrial blood pressure (*A*) and left atrial-tracheal pressure relationship (*B*)**
Abbreviations: LA, left atrial; other abbreviations as in Fig. 1. Results in panel *A* are displayed using the same format and time points defined in Fig. 1. Data are expressed as means(SD); *n* = 10 for non-asphyxial and asphyxial ECC groups, *n* = 11 for DCC group. [a]*P* = 0.002, [b]*P* < 0.001, compared to preceding time point. Linear regression relationship between mean tracheal and LA pressures using combined data points at fetal baseline, after reduction of lung liquid volume and following fetal delivery in the non-asphyxial ECC (open circles), asphyxial ECC (open triangles), and DCC (filled squares) groups. Note that the slopes of individual relationships in these groups are not different (*P* = 0.771).

cord clamping (DCC), has produced three main findings. First, the non-ventilatory events of a reduction in lung liquid volume, complete delivery of the fetus from the uterus and clamping of the umbilical cord all increased PA blood flow, with their cumulative effect contributing ∼30% of the perinatal rise in PA blood flow between the baseline fetal state and early neonatal peak after ECC or DCC. Second, these non-ventilatory increases in PA blood flow were rapidly negated after ECC if an asphyxial level of arterial oxygenation developed prior to ventilation. Third, different physiological mechanisms with distinct temporal patterns supported non-ventilatory and ventilatory rises in PA flow during the birth transition.

Although reduction of lung liquid volume and complete delivery of fetal lambs produced stepwise rises in PA blood flow (Fig. 3*A*), pulmonary vascular conductance (Fig. 3*B*) and right ventricular (RV) outflow (Fig. 4*A*), these interventions also progressively decreased mean left atrial blood pressure (Fig. 2*A*). That these blood flow and pressure changes were primarily related to mechanical effects arising from decreased constraint of the pulmonary microvasculature and the heart was consistent with four observations. First, reducing lung liquid volume decreases external compression of small pulmonary arteries in fetal lungs (Walker et al., 1988). Second, mean left atrial

and tracheal pressures were linearly related between the baseline state and delivery of the fetus (Fig. 2*B*), implying that measured left atrial blood pressure included a pressure component transmitted from the fluid-filled lungs into the thin-walled left atrial chamber (Smolich, 2014; Smolich & Mynard, 2019; Smolich, Cheung et al., 2021). Third, the fluid-filled lungs impose a physical constraint on the heart which restricts its dimensions (Grant et al., 1992; Smolich, Cheung et al., 2021) and limits its pumping performance (Grant & Walker, 1996; Grant et al., 1992; Smolich, Cheung et al., 2021), with diminution of this constraint increasing fetal cardiac output via the Frank-Starling mechanism (Grant, 1999; Grant et al., 1992; Smolich, Cheung et al., 2021). Fourth, delivery of the chest after labour at birth increases ventricular end-diastolic diameter of the chronically instrumented lamb (Kirkpatrick et al., 1973). However, the genesis of rises in PA flow and RV output, as well as a fall in left atrial pressure, probably differed between reduction of lung liquid volume and fetal delivery in the current study, as the presence of a clamped endotracheal tube during fetal delivery prevented any physical decrease in lung liquid volume. Rises in blood flows and a fall in left atrial pressure after fetal delivery were therefore likely to be mainly related to loss of thoracic constraint arising

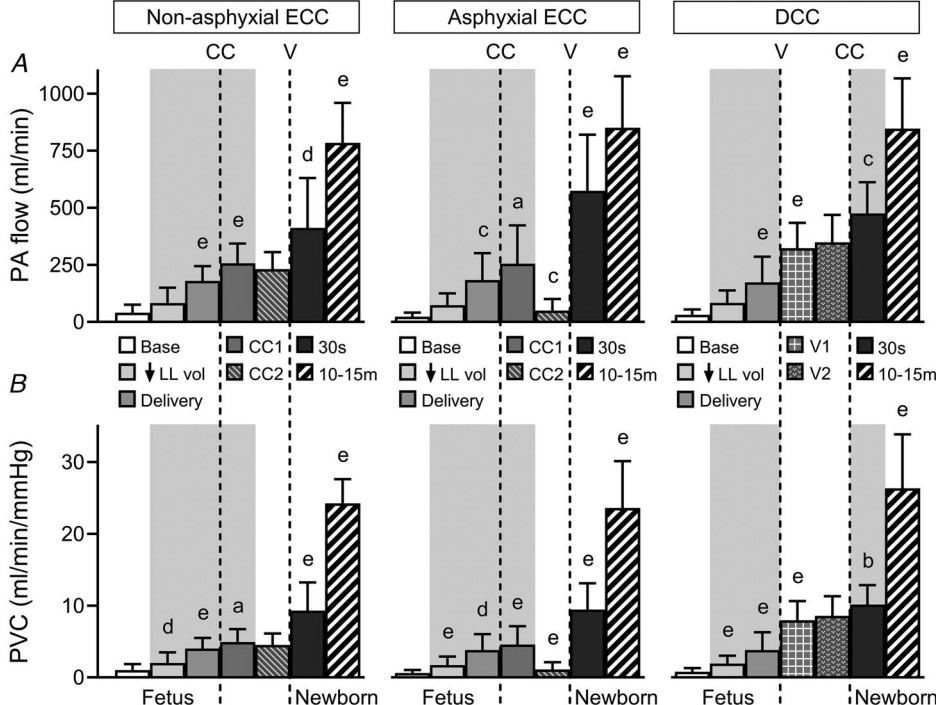

**Figure 3. Pulmonary arterial blood flow (*A*) and pulmonary vascular conductance (*B*)**
Abbreviations: PA, pulmonary arterial; PVC, pulmonary vascular conductance; other abbreviations as in Fig. 1. Results are displayed using the same format and time points defined in Fig. 1. Note that shaded regions overlie non-ventilatory events of the birth transition, Data are expressed as means(SD); $n = 10$ for non-asphyxial and asphyxial ECC groups, $n = 11$ for DCC group. [a]$P \leq 0.035$, [b]$P = 0.033$, [c]$P \leq 0.013$, [d]$P \leq 0.006$, [e]$P \leq 0.001$, compared to preceding time point.

from surrounding amniotic fluid and maternal abdominal contents (Grant, 1999; Grant et al., 1992).

On the basis of phasic flow analysis, two distinct phenomena underpinned increases in mean PA blood flow occurring with reduction of lung liquid volume and fetal delivery. The first was a rise in PA systolic inflow (Fig. 4*B*) that was a direct consequence of a greater RV output (Fig. 4*A*), as the proportional distribution of RV outflow into PA systolic inflow was unchanged (Fig. 4*C*). The second phenomenon reflected an alteration in a characteristic dynamic feature of fetal pulmonary perfusion, namely that a portion of PA systolic inflow does not traverse the lungs, but instead exits distensible large PA vessels in late-systole and diastole as a back-flow that is the main source of continuous diastolic right-to-left flow across the ductus arteriosus (Rudolph, 2009b; Smolich, 2014; Smolich et al., 2012a). Thus, while ∼80% of PA systolic inflow returned as a backflow in the baseline fetal state, this proportion fell stepwise to ∼60% with a reduction in lung liquid volume and then to ∼30% with fetal delivery (Fig. 5*B*). These decreases had a substantial impact on pulmonary perfusion as comparison of sequential decrements in backflow (27 and 63 ml min$^{-1}$, Fig. 5*A*) and increments in mean PA flow (48 and 99 ml min$^{-1}$, Fig. 3*A*) occurring with reduction of lung liquid volume and fetal delivery, respectively, suggested that decreased PA backflow contributed ∼55% and ∼65% of these rises in mean PA flow. Furthermore,

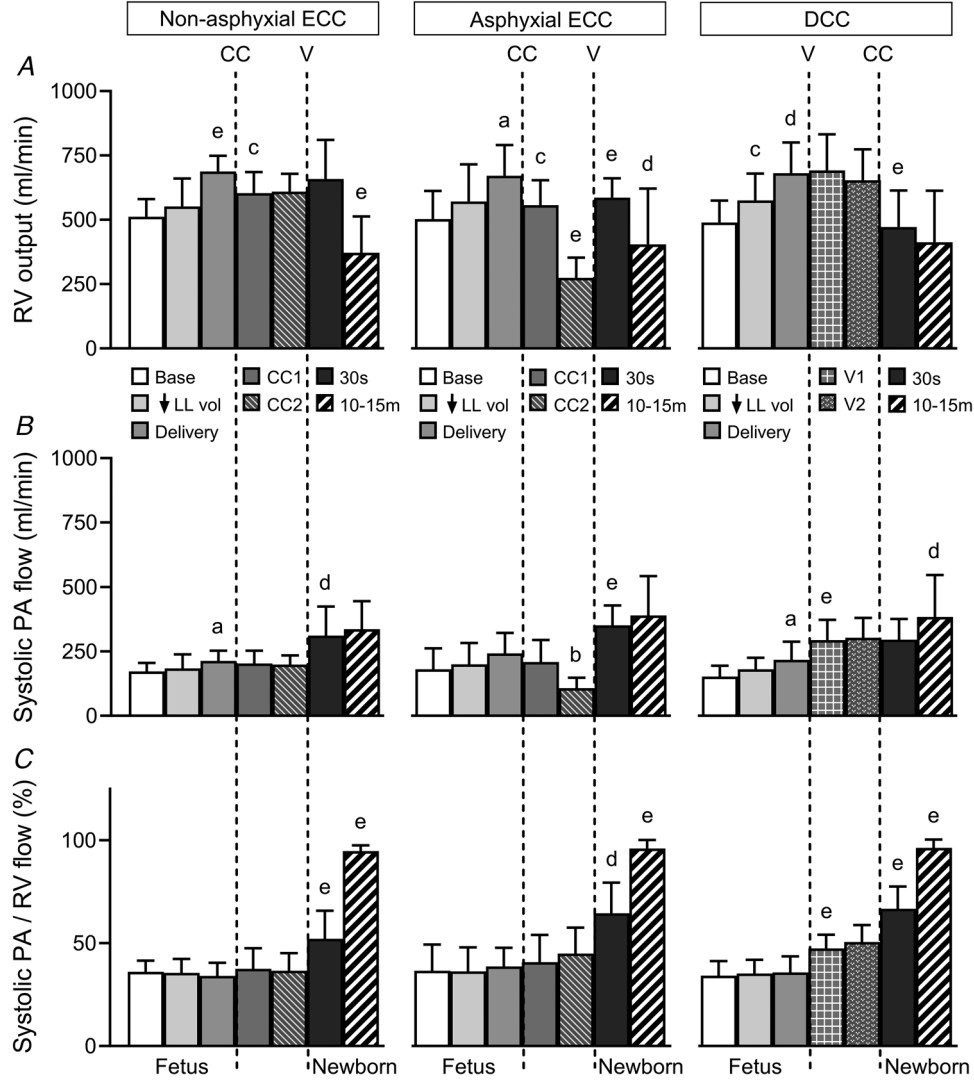

**Figure 4. Right ventricular output (*A*), systolic pulmonary arterial blood flow (*B*) and percentage distribution of right ventricular systolic outflow into pulmonary arterial systolic inflow (*C*)**
Abbreviations: PA, pulmonary arterial; RV, right ventricular; other abbreviations as in Fig. 1. Results are displayed using the same format and time points defined in Fig. 1. Data are expressed as means(SD); $n = 10$ for non-asphyxial and asphyxial ECC groups, $n = 11$ for DCC group. [a]$P \leq 0.049$, [b]$P < 0.021$, [c]$P \leq 0.012$, [d]$P \leq 0.005$, [e]$P < 0.001$, compared to preceding time point.

the presence of a highly significant negative linear relationship between pulmonary vascular conductance and the PA backflow-to-inflow ratio (Fig. 5*C*) implied that an increasing vascular cross-sectional area related to pulmonary vasodilatation and/or vascular recruitment was a major factor underlying falls in PA backflow.

The proposition that a reduction of lung liquid volume increases pulmonary blood flow in the fetus is well

established experimentally (Hooper, 1998; Smolich, 2014; Smolich & Mynard, 2019; Walker et al., 1988). On the other hand, the finding that PA blood flow rose with complete delivery of the fetus from the uterus (Fig. 3*A*) runs counter to a view that pulmonary blood flow is unaffected by fetal delivery, based on an illustrative figure depicting relatively minor changes in the pulse width and peak of the left PA blood flow signal after

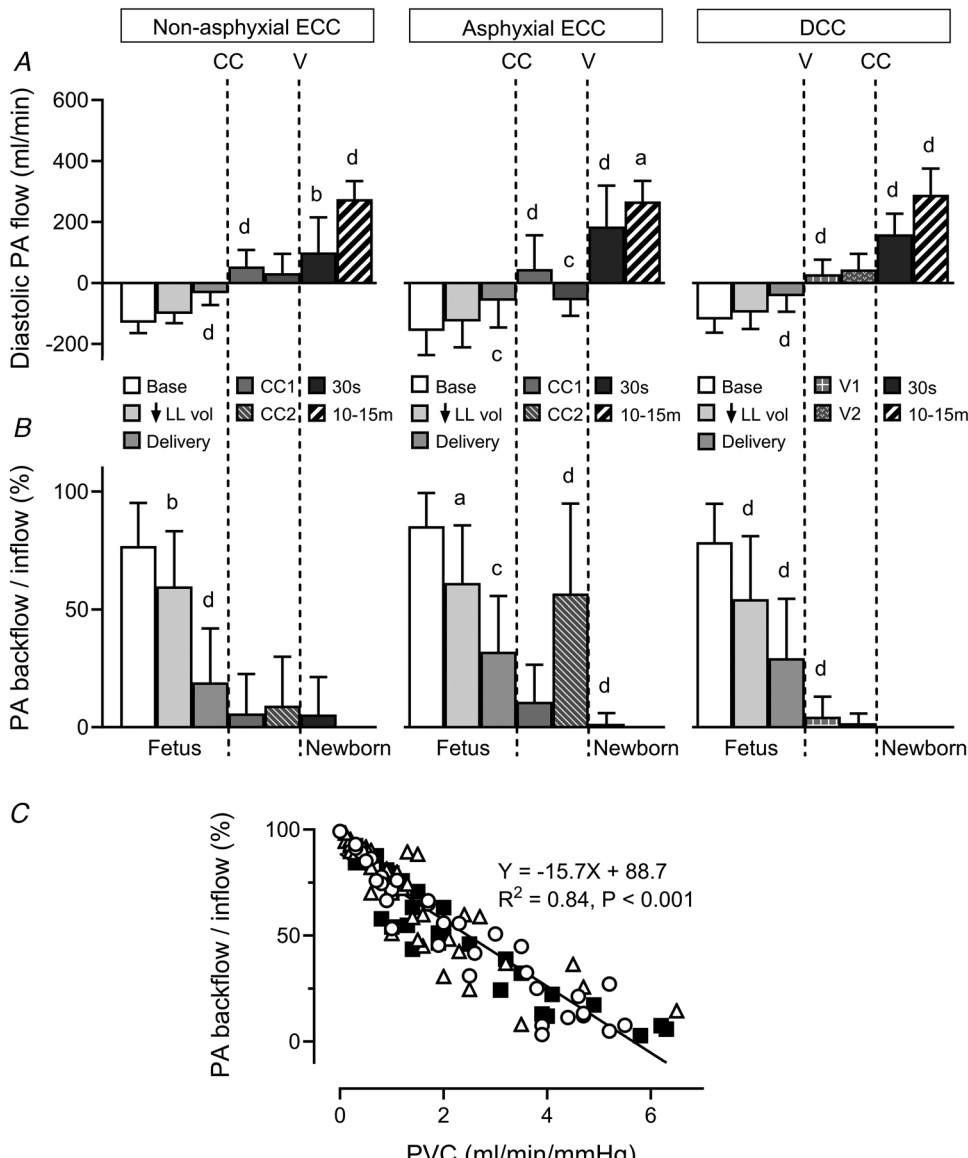

**Figure 5. Diastolic pulmonary arterial blood flow (*A*), pulmonary arterial backflow-to-inflow ratio (*B*), and the relationship between pulmonary arterial backflow-to-inflow ratio and pulmonary vascular conductance (*C*)**
Abbreviations: PA, pulmonary arterial; PVC, pulmonary vascular conductance; other abbreviations as in Fig. 1. Results in *A* and *B* are displayed using the same format and time points defined in Fig. 1. Data are expressed as means (SD); *n* = 10 for non-asphyxial and asphyxial ECC groups, *n* = 11 for DCC group. [a]$P \leq 0.036$, [b]$P \leq 0.024$, [c]$P \leq 0.008$, [d]$P < 0.001$, compared to preceding time point. Linear regression relationship between PA backflow/inflow ratio and PVC using combined datapoints from fetal baseline, after reduction of lung liquid volume and following fetal delivery in the non-asphyxial ECC (open circles), asphyxial ECC (open triangles), and DCC (filled squares) groups. Note that the slopes of individual relationships in these groups are not different (*P* = 0.599).

delivery of a fetal lamb prior to ECC (Polglase et al., 2005). However, comparison of percentage changes in maximal $(+3(14)\%)$, pulse $(-4(14)\%)$ and mean PA flow $(+185(179)\%)$ occurring with fetal delivery in our combined study group $(n = 31)$ suggested that changes in either the pulse width or peak of the PA blood flow signal did not accurately reflect alterations in mean PA flow at delivery. A plausible basis for much of this divergence is that, as detailed above, most ($\sim$65%) of the increase in mean PA flow occurring with fetal delivery results from reduced PA backflow, which is manifested as a less negative diastolic offset within the body of the PA flow signal, rather than an alteration in pulse width or peak.

PA blood flow also rose with clamping of the umbilical cord (Fig. 3A), not only in the initial period after ECC (Smolich & Kenna, 2022; Smolich et al., 2015), but also after DCC following prior ventilation (Bhatt et al., 2013; Smolich & Kenna, 2022; Smolich & Kenna, 2025; Smolich et al., 2024). Furthermore, this increase in PA flow was accompanied by a rise in pulmonary vascular conductance (Fig. 3B), implying involvement of pulmonary vaso-dilatation and/or increased vessel recruitment. However, differences were evident between ECC and DCC in the processes supporting these increases. Thus, while RV output fell with cord clamping in both ECC and DCC,

the associated proportion of RV output distributed as systolic PA flow was unchanged in ECC but increased in DCC (Fig. 4C). Moreover, cord clamping resulted in loss of PA backflow in ECC, but an increase in forward PA diastolic flow after DCC (Fig. 5A). On the other hand, an increase in left-to-right ductal shunting after cord clamping, which mainly originates as diastolic discharge from a lower body arterial reservoir (Smolich & Kenna, 2025; Smolich et al., 2016, 2020), was evident after both ECC and DCC (Fig. 6A).

Although the specific mechanism(s) underlying increases in PA blood flow and pulmonary vascular conductance after cord clamping at birth are yet to be defined experimentally, at least two processes are likely to be involved. First, a major factor limiting the level of PA flow in the fetus is a decreased pulmonary microvascular cross-sectional area secondary to external constraint arising from lung liquid in the future airways and alveoli (Walker et al., 1988). An increase in PA blood pressure that occurs with cord clamping (Bhatt et al., 2013; Smolich & Kenna, 2022; Smolich et al., 2015) would oppose this constraint, thereby facilitating a rise in pulmonary microvascular cross-sectional area, and thus PA blood flow and pulmonary vascular conductance. Second, the association of a higher PA blood flow after cord clamping

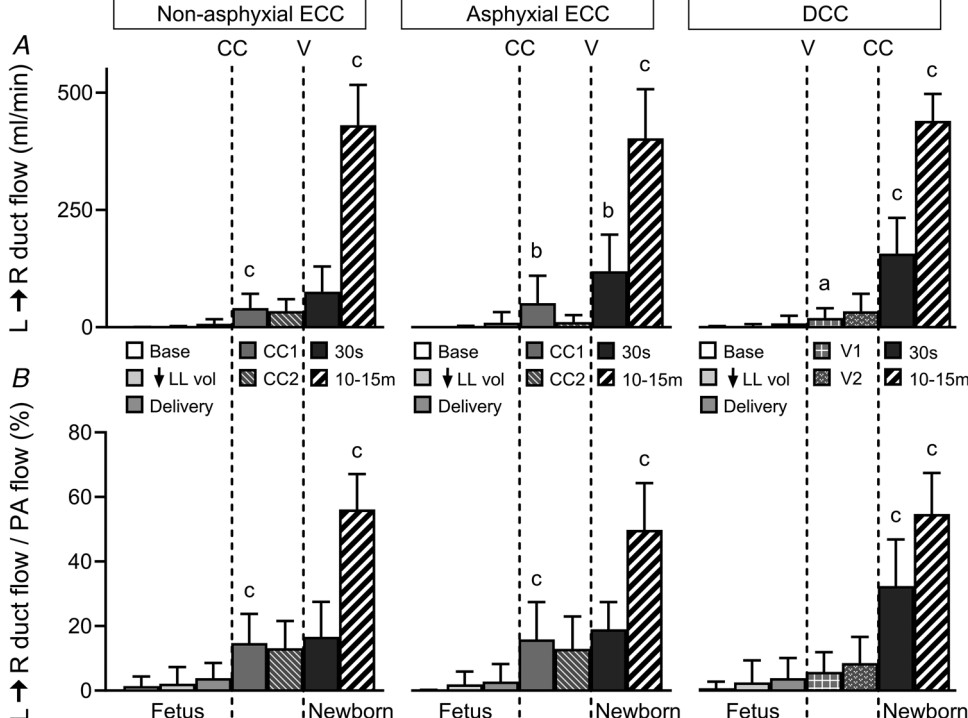

**Figure 6. Left-to-right ductal blood flow (*A*) and the percentage contribution of left-to-right ductal flow to pulmonary arterial blood flow (*B*)**
Abbreviations: L→R, left-to-right; PA, pulmonary arterial; other abbreviations as in Fig. 1. Results are displayed using the same format and time points defined in Fig. 1. Data are expressed as means (SD); $n = 10$ for non-asphyxial and asphyxial ECC groups, $n = 11$ for DCC group. [a]$P = 0.011$, [b]$P \leq 0.002$, [c]$P < 0.001$, compared to preceding time point.

with increases in pulmonary vascular conductance and pulmonary trunk blood pressure (Figs 1*B* and 3*B*) resembles the pattern of an initial rise in PA blood flow produced by partial constriction of the ductus arteriosus in the fetus, which is considered to be related to an increase in PA wall stress that leads to rapid activation of endothelium-dependent vasodilatation (Abman & Accurso, 1989; Ovadia et al., 2002; Parker et al., 2005).

As reported previously (Smolich et al., 2015, 2020), PA blood flow decreases rapidly if arterial oxygenation falls to an asphyxial level during an extended interval (>1 min) prior to ventilation after ECC, with this flow attaining a value comparable to the baseline fetal state (Fig. 3*A*), in conjunction with a large reduction of pulmonary vascular conductance (Fig. 3*B*) and a return of substantial of PA diastolic backflow (Fig. 5*A* and *B*). Importantly, however, the basis of a low PA flow after ECC was quite different from the physiological baseline fetal state, as it represented the net result of two opposing effects.

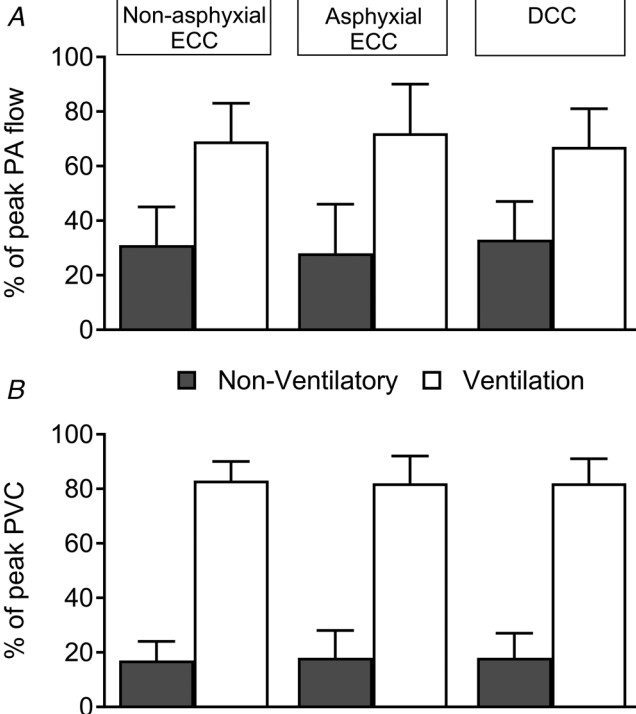

**Figure 7. Percentage of peak neonatal pulmonary arterial blood flow (*A*) and pulmonary vascular conductance (*B*) arising from non-ventilatory events and ventilation**
Abbreviations: PA, pulmonary arterial; PVC, pulmonary vascular conductance; DCC, delayed cord clamping; ECC, early cord clamping. Note that the total fractional contribution of non-ventilatory events to the rise in PA flow or PVC between the fetal baseline and the peak nowborn value at 10–15 min after birth has been obtained by summing within-animal increments related to each non-ventilatory event in Fig. 3. Data are expressed as means(SD); *n* = 10 for non-asphyxial and asphyxial ECC groups, *n* = 11 for DCC group.

These comprised an intense pulmonary vasoconstriction activated by markedly diminished arterial oxygenation (Peeters et al., 1979; Rudolph, 2009*b*), which negated the preceding pulmonary vasodilatation and rise in PA flow resulting from the cumulative effect of a reduction in lung liquid volume, fetal delivery and umbilical cord clamping (Fig. 3).

PA blood flow rose substantially after ventilation with either ECC or DCC (Fig. 3*A*), with the absolute and relative distribution of RV output into systolic PA flow increased in both scenarios (Fig. 4). However, an increase in left-to-right ductal shunting (Fig. 6*A*) and an associated positive PA diastolic flow (Fig. 5*A*) was more pronounced with ECC. Moreover, the magnitude of an increase in PA flow with ECC was also dependent on the cord clamp-to-ventilation interval. Thus, while PA flow fell markedly in the presence of very low arterial oxygenation after ECC, the rise in this flow induced by ventilation was then more pronounced, with absolute levels of PA flow similar in the non-asphyxial and asphyxial ECC groups within 30 s after onset of ventilation (Fig. 3*A*). However, the downside of a larger increment in PA flow with onset of ventilation in the asphyxial ECC group was that it occurred in conjunction with overshoots of arterial pressures, heart rate and cardiac contractility that resulted from marked asphyxia-induced elevations in circulating concentrations of the catecholamines noradrenaline and adrenaline (Smolich et al., 2017).

A striking finding of the present study was that, despite differences in the initial pattern of haemodynamic and PA blood flow changes occurring with cord clamping and ventilation in the three groups, peak levels of this flow at 10–15 min after birth were not only quite similar (Fig. 3*A*), but were also supported by similar levels of systolic PA inflow and PA distribution of RV output (Fig. 4), diastolic PA forward flow (Fig. 5*A*) and left-to-right shunting across the ductus (Fig. 6*A*). Moreover, even with a pronounced decrease in PA flow and pulmonary vascular conductance related to a marked post-cord clamping fall in arterial oxygenation in the asphyxial ECC group (Fig. 3), the overall contribution of ventilation and non-ventilatory events to peak PA flow and PVC in the three groups after birth were also similar (Fig. 7).

Viewed together, the findings of the current study are in accord with the proposition that the physiological mechanisms which support a rising PA blood flow during the birth transition have a defined temporal pattern. Thus, non-ventilatory increases in PA flow were predominantly related to effects arising from reduced constraint of the pulmonary microvasculature and heart, which increased RV output (Fig. 4*A*) and PA systolic inflow (Fig. 4*B*), but decreased PA backflow (Fig. 5). On the other hand, ventilatory increases in PA flow were mainly related to a more pronounced distribution of RV output to the lungs

(Fig. 4C) and greater left-to-right shunting across the ductus arteriosus (Fig. 6).

The results of the current study do not support the view that an increase in pulmonary perfusion at birth is not triggered until the onset of lung aeration (Hooper, Polglase, & Roehr, 2015; Hooper, Polglase, & te Pas, 2015; Hooper, Te Pas et al., 2015). This view is largely reliant on conclusions reached from synchrontron-based X-ray studies in rabbit kittens (Lang et al., 2016, 2017), where the observed pattern of changes in pulmonary blood flow was a direct consequence of two specific features of the experimental design. First, initial baseline pulmonary blood flow measurements were not performed until just before the start of ventilation (Lang et al., 2016, 2017), which thus precluded any assessment of the potential contributions of a fall in lung liquid volume, fetal delivery and ligation/cutting of the umbilical cord to changes in this flow. Second, during a 2–3 min interval between ligation/cutting of the umbilical cord and the start of ventilation, kittens developed a profound bradycardia of 47–69 beats $min^{-1}$ (Lang et al., 2016, 2017), which is markedly lower than a normal resting heart rate of $\sim$230—250 beats $min^{-1}$ in late-gestation fetal rabbits (Coombs et al., 2020; Turna & Erdogan, 2016). This bradycardia is consistent with the presence of an asphyxial level of arterial oxygenation (Godfrey, 1968) which, on the basis of fetal lamb data, rapidly develops within $\sim$1 min after ECC in the absence of ventilation (Smolich et al., 2015, 2017). Reduced arterial oxygenation is a potent stimulus for intense pulmonary vasoconstriction and an associated marked decrease of pulmonary blood flow in the fetus (Campbell et al., 1967; Peeters et al., 1979; Rudolph, 2009b; Smolich et al., 2015, 2020), with relief of this low oxygenation state after the onset of ventilation resulting in a surge of PA blood flow (Smolich et al., 2015, 2020).

## Limitations

The present study has three main potential limitations. The first is that experiments were performed in an acutely instrumented and anaesthetized preparation without surgical closure of a left thoracotomy, an approach necessary because of the extent of instrumentation required to obtain high-fidelity blood flow and pressure measurements at multiple intrathoracic sites. However, as noted previously (Smolich & Kenna, 2022, 2025; Smolich & Mynard, 2019; Smolich et al., 2017, 2020, 2024; Smolich, Cheung et al., 2021; Smolich, Kenna et al., 2021), key features of the fetal baseline state and birth transition in this acute preparation closely resembled those of chronically instrumental preterm fetal lambs (Bhatt et al., 2013; Crossley, Allison et al., 2009; Crossley Morley et al., 2009). In particular, PA blood flow

at the fetal baseline (32 ml $min^{-1}$, corresponding to 9 ml $min^{-1}$ (kg body weight)$^{-1}$) and its proportion of RV output (6%) were within the range of characteristically low values obtained in chronically instrumented fetal lambs (Heymann, 1999; Rudolph, 2009a).

The second limitation is that substantial differences exist between human births and experiments in lambs, which may limit clinical translation of study findings. Thus, (1) experiments related solely to Caesarean section preterm birth without labour, (2) lambs were intubated and received positive-pressure mechanical ventilation, with spontaneous ventilation suppressed by general anaesthesia, (3) pressures employed during mechanical ventilation of lambs were higher than would be used for human infants, with a relatively high positive end-expiratory pressure of 8 $cmH_2O$ physiologically appropriate to optimize dynamic compliance of lungs in preterm lambs (Tingay et al., 2014).

The third limitation is that the experimental design consisted of sequential and discrete changes in non-ventilatory events during the birth transition whereas, in a clinical setting, these events may be concurrent, e.g. loss of lung liquid through an open glottis at the time of fetal delivery (Hooper, Polglase, & Roehr, 2015; Hooper et al., 2019). In addition, lung liquid volume was reduced solely by physical drainage, whereas a decrease in lung liquid volume during labour and a vaginal delivery may result not only from uptake of lung liquid into the lung interstitium secondary to elevations in the concentration of circulating adrenaline, but also mechanical expulsion of lung liquid via the trachea *in utero*, and loss of lung liquid via an open glottis after delivery (Berger et al., 1998; Brown et al., 1983; Hooper, Polglase, & Roehr, 2015; Stockx et al., 2007).

## Clinical implications

The results of the present study have two main potential clinical implications for pulmonary perfusion in the birth transition. First, increases in fetal PA blood flow related to non-ventilatory events in the birth transition are rapidly negated if pulmonary vasoconstriction supervenes secondary to very low levels of arterial oxygenation after ECC, as evident in Fig. 3A and (Smolich et al., 2015, 2020). From the viewpoint of enhancing PA blood flow and then maintaining this increase during the birth process and after delivery, this scenario highlights the importance of not only preventing/minimizing the effect of factors which can reduce arterial oxygenation prior to onset of ventilation, e.g. torsion, kinking, spasm or tension of the umbilical cord, but if ECC is employed, also ensuring that effective ventilation after delivery occurs within the first 'golden minute' (Wyckoff et al., 2015; Wyllie et al., 2015). Although the level of PA flow is rapidly restored after

the onset of ventilation in the event of such preceding pulmonary vasoconstriction (Fig. 3*A*), this restoration occurs in conjunction with marked haemodynamic fluctuations related to catecholamine-induced surges in heart rate, arterial blood pressures and cardiac contractility (Smolich et al., 2017).

Second, preservation of non-ventilatory increases in pulmonary blood flow may be an important benefit of employing DCC rather than ECC at birth, as experimental studies have demonstrated that DCC avoids very rapid declines in arterial oxygenation evident between ECC and subsequent ventilation (Polglase et al., 2015; Smolich & Kenna, 2022; Smolich et al., 2015, 2017, 2020; Smolich, Kenna, Mynard, Phillips et al., 2019; Smolich, Kenna et al., 2021), while both experimental (Polglase et al., 2015; Smolich & Kenna, 2022; Smolich, Kenna, Mynard, Phillips et al., 2019) and clinical studies (Kc et al., 2019; Padilla-Sanchez et al., 2020; Smit et al., 2014) have demonstrated that levels of arterial oxygenation in the first few minutes after birth are higher with DCC than ECC.

## Conclusions

The findings of this study suggest that lung aeration following the onset of ventilation is a major, but not sole, driver of increases in pulmonary perfusion during the pre-term birth transition, as the non-ventilatory events of a reduction in lung liquid volume, complete delivery of the fetus from the uterus and cord clamping provide a substantial contribution (∼30%) to perinatal increases in PA flow in the setting of both ECC and DCC. However, this non-ventilatory contribution to PA flow is negated if an asphyxial level of arterial oxygenation develops after ECC. Moreover, the physiological mechanisms supporting perinatal rises in PA flow undergo a progressive shift with a defined temporal pattern.

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

## Additional information

### Data availability statement

Data used for figures and analyses are available upon request to the corresponding author from 6 months following article

publication to researchers who provide a methodologically sound proposal, with approval by an independent review committee ('learned intermediary'). Proposals should be directed to joe.smolich@mcri.edu.au to gain data access. Data requestors will need to sign a data access or material transfer agreement approved by MCRI.

## Competing interests

The author has no competing interests, financial or otherwise, in respect of this study.

## Author contributions

Sole author.

## Funding

This work was supported by a Project Grant (1 105 137) from the National Health and Medical Research Council of Australia (NHMRC), philanthropic funds from the Heart Research Group and the Victorian Government's Operational Infrastructure Support Program.

## Acknowledgements

Experimental studies were performed within the Translational Research Unit of the Murdoch Children's Research Institute. Thanks are extended to Magdy Sourial, Rebecca Sutton, Amy Tilley, and Sara White for their assistance with experimental studies, to Dr Kelly Kenna for assistance with experimental studies and manuscript preparation, to Associate/Professor Jonathan Mynard for development of data acquisition and analysis software, and to Professor Salvatore Pepe for review of the manuscript.

Open access publishing facilitated by The University of Melbourne, as part of the Wiley - The University of Melbourne agreement via the Council of Australian University Librarians.

## Keywords

birth transition, delayed cord clamping, early cord clamping, fetal delivery, lung liquid volume, pulmonary arterial blood flow

## Supporting information

Additional supporting information can be found online in the Supporting Information section at the end of the HTML view of the article. Supporting information files available:

**Peer Review History**

