## [Peer Review History · The Journal of Physiology]

Non-ventilatory events substantially contribute to rise in pulmonary arterial blood flow with early or delayed cord clamping at birth in preterm lambs

Joseph John Smolich
DOI: 10.1113/JP288749

Corresponding author(s): Joseph Smolich (joe.smolich@mcri.edu.au)

Review Timeline:

Submission Date:	16-Feb-2025
Editorial Decision:	10-Apr-2025
Revision Received:	03-Jun-2025
Editorial Decision:	23-Jun-2025
Revision Received:	06-Jul-2025
Accepted:	11-Jul-2025

Senior Editor: Laura Bennet

Reviewing Editor: Christopher Lear

Transaction Report:

Dear Dr Smolich,

Re: JP-RP-2025-288749 "Non-ventilatory events substantially contribute to rise in pulmonary arterial blood flow with early or delayed cord clamping at birth in preterm lambs" by Joseph John Smolich

Thank you for submitting your manuscript to The Journal of Physiology. It has been assessed by a Reviewing Editor and by 2 expert referees and we are pleased to tell you that it is acceptable for publication following satisfactory revision.

REVISION CHECKLIST:

We look forward to receiving your revised submission.

Yours sincerely,

Laura Bennet
Senior Editor
The Journal of Physiology

REQUIRED ITEMS

- Author photo and profile. First or joint first authors are asked to provide a short biography (no more than 100 words for one author or 150 words in total for joint first authors) and a portrait photograph. These should be uploaded and clearly labelled together in a Word document with the revised version of the manuscript. See Information for Authors for further details.

- You must start the Methods section with a paragraph headed Ethical approval (https://jp.msubmit.net/cgi-bin/main.plex?form_type=display_requirements#methods).

Research must comply with The Journal's policies regarding animal experiments (<https://physoc.onlinelibrary.wiley.com/hub/animal-experiments>) and adherence to these policies must be stated in the manuscript.

Authors should confirm in their Methods section that their experiments were carried out according to the guidelines laid down by their institution's animal welfare committee, including an ethics approval reference number. The Methods section must contain a statement about access to food, water and housing, details of the anaesthetic regime: anaesthetic used, dose and route of administration, and method of killing the experimental animals.

- Please upload separate high-quality figure files via the submission form.

- Please ensure that the Article File you upload is a Word file.

- Please include an Abstract Figure file, as well as the Figure Legend text within the main article file. The Abstract Figure is a piece of artwork designed to give readers an immediate understanding of the research and should summarise the main conclusions. If possible, the image should be easily 'readable' from left to right or top to bottom. It should show the physiological relevance of the manuscript so readers can assess the importance and content of its findings. Abstract Figures should not merely recapitulate other figures in the manuscript. Please try to keep the diagram as simple as possible and without superfluous information that may distract from the main conclusion(s). Abstract Figures must be provided by authors no later than the revised manuscript stage and should be uploaded as a separate file during online submission labelled as File Type 'Abstract Figure'. Please also ensure that you include the figure legend in the main article file. All Abstract Figures should be created using BioRender. Authors should use The Journal's premium BioRender account to export high-resolution images. Details on how to use and access the premium account are included as part of this email.

EDITOR COMMENTS

Reviewing Editor:

Thank you for your manuscript detailing the contributors to increasing pulmonary blood flow at birth. The reviewers have noted the importance of these findings but have several suggestions for improvement. In particular please address the statistics suggestion from reviewer 2 which would likely strengthen and improve the impact of the present findings.

I would like to request the author attempts to improve the presentation of the figures. I appreciate this is a difficult dataset to present, but at present I found the figures difficult to follow and not very engaging. Some thoughts for the author to consider:

- 1 - Although this is not continuous data in even epochs, as it is a timecourse and needs to be viewed as such, I would suggest line graphs rather than boxes would be easier to follow.
- 2 - The horizontal stacking of each group also makes the data hard to follow, and the change in the x axis labelling can on the third can be easily missed. I would suggest the two ECC groups could be plotted on the same x axis, with the DCC group on its separate x axis (accepting that the sequence of events is different). Even 2 vs 3 graphs would make it easier to follow. Ability to compare groups would be enhanced too, even if that is not a key goal of this analysis.
- 3 - Please attempt to find a clearer manner to label sequence of events on the x axes. Shading etc as per the authors previous work on the same model would be preferable.
- 4 - Please make clear if food/water was needed/provided as these are acute experiments. Please confirm access while animals were being housed prior to experiments.

Senior Editor:

Thank you for submission. Please address all the comments raised by the reviewers, and in particular those related to alternate ways to better display and analyse the data. All reviewers have raised the need to clarify various aspects regarding the data. While these are acute experiments, please provide housing and care information if animals were held in your unit prior to the experiments as per The Journal's guidelines

REFeree COMMENTS

Referee #1:

Smolich presents an elegant series of studies which evaluate the contribution of reduction in lung volume, delivery and cord clamping, as well as ventilation, on changes in central hemodynamics, and most notably pulmonary blood flow in an acute preterm lamb surgical preparation. The investigator uses a well-established model for this studies. Data are presented clearly and extend our understanding of hemodynamic changes associate with birth. There are a few points that require clarification.

1. Was maternal administration of glucocorticoids used? For the delivery of most preterm infants, antenatal glucocorticoids, which impact pulmonary function are provided. Though mainly provide to promote surfactant production, this does appear to affect lung water through ENaC.
2. The timing of the AoT blood sampling in ECCNA and ECCA are confusing. 20 seconds after clamping in ECCNA with ventilation at 36 sec, and 30 seconds after clamping in ECCA with ventilation at 100 sec. Why so soon in ECCA relative to ventilation time? Thus, blood gas may not be entirely reflective of PaO₂ at time of ventilation in ECCA group - but explains the relatively small differences in blood gas values between groups (Table 2).
3. The ventilatory settings (or at least upper limits thereof) are much higher than would typically be used for preterm infants. I'm curious as to why this was necessary as such high ventilator pressures, particularly in a setting of an open chest, likely impact pulmonary vascular resistance and thus central hemodynamics.
4. The hemodynamic recording periods are relatively short (20-30 seconds, though necessitated by the design of the study). How much intra-animal variability was there in the measurements of various blood flows during this period?
5. Were there animal preparations for which data were not included? If so, how many and why?
6. Consider softening the comment regarding the data refuting the view that an increase in pulmonary perfusion.....lung aeration (lines 498-99). In reviewing these references, though the authors conclude ventilation is the primary stimulus for increase in blood flow, I don't believe they comment about an absence of increase in pulmonary blood flow until aeration. They also demonstrate that aeration is not needed to increase blood flow as they demonstrate, in their model, that blood flow increases in areas of the lung not aerated (or expanded). I recognize Smolich studied an immediate period of time after delivery (and before) that Lang and Hooper did not.
7. The author appropriate acknowledges limitations of his model. In listing these (lines 520-22), please include open-chested and mechanically ventilated with positive pressure for completeness. It should also be restated that the experiments were performed in preterm animals in the absence of the effect of labor and natural decrease in lung liquid volume that occurs as

the fetus approaches term.

8. I appreciate the clinical implication comment regarding earlier vs later initiation of ventilation to enhance PBF. What may be of interest, and perhaps unknown (though not likely) to the investigator, is that two large trials of early ventilation/resuscitation in preterm infants failed to have an effect on study outcomes (though PBF was obviously not assessed, including intraventricular hemorrhage. (PMID 39671198, 38758557). Given the data regarding improvements in hemodynamic fluctuations shown in the present study and others by the investigator by early ventilation, these findings were very disappointing.

Referee #2:

This study used acutely-instrumented preterm lambs to quantify the contribution of a reduction in lung liquid volume, fetal delivery and cord clamping, as well as ventilation to the increase in pulmonary blood flow at birth.

These interesting results confirm this respondent's bias that ventilation is the major factor, but provide important and novel insight by quantifying a roughly 30% contribution from multiple other factors.

This complex study is performed appropriately, using well validated methods.

The major limitation is the use of anaesthesia, which abolishes reflex pathways. The author acknowledges and carefully discusses this important issue.

Ibid. Please include 'anaesthetised' in the title and abstract.

It isn't clear to this non-statistician how the contributions of each factor were quantified. I think it is based on crude means at different times. It has been suggested that a mixed model analysis can be more robust in this setting? I respectfully suggest that obtaining professional statistical input may be helpful for such a complex, multifactorial analysis of longitudinal data, and that a more refined analysis can only improve the impact of this very interesting ms.

Minor.

Figures; please replace bar graphs with vertical scatter plots, with median or mean bars.

Trivia.

there are too many nonstandard abbreviations that make the MS hard to follow. DCC means something different to this respondent for example. ECC and DCC in particular should be spelt out please, as should nearly all the remainder.

"with most of this output" unclear. What output? Best guess that you mean of total RV output here

"complete delivery of the fetus from the uterus, which augments pulmonary arterial (PA) blood flow and RV output," how is this likely to work? Suggest briefly suggest the likely mechanism here, to lead in to the detailed discussion.

"ensemble-averaged". What does this mean specifically in this context? There is no discussion of statistical models to be averaged.

The author may wish to review some emotive language in the results... "then surged".

END OF COMMENTS

Reviewing Editor

Thank you for your manuscript detailing the contributors to increasing pulmonary blood flow at birth. The reviewers have noted the importance of these findings but have several suggestions for improvement. In particular please address the statistics suggestion from reviewer 2 which would likely strengthen and improve the impact of the present findings.

Thank you for these comments. The suggestion of Reviewer 2 regarding statistical analysis of the data has been addressed in detail below.

I would like to request the author attempts to improve the presentation of the figures. I appreciate this is a difficult dataset to present, but at present I found the figures difficult to follow and not very engaging. Some thoughts for the author to consider:

1 - Although this is not continuous data in even epochs, as it is a timecourse and needs to be viewed as such, I would suggest line graphs rather than boxes would be easier to follow.

2 - The horizontal stacking of each group also makes the data hard to follow, and the change in the x axis labelling can on the third can be easily missed. I would suggest the two ECC groups could be plotted on the same x axis, with the DCC group on its separate x axis (accepting that the sequence of events is different). Even 2 vs 3 graphs would make it easier to follow. Ability to compare groups would be enhanced too, even if that is not a key goal of this analysis.

3 - Please attempt to find a clearer manner to label sequence of events on the x axes. Shading etc as per the authors previous work on the same model would be preferable.

Yes, it has been a difficult dataset to present in a format that easily conveys the main messages of the study. In respect of the specific suggestion made, I would prefer not to use a line graph representation because, as the Editor acknowledges, data are not continuous or in even epochs, so such a presentation might be potentially misleading and/or confusing for some readers.

After a rethink about data presentation and consideration of the pros and cons of various options suggested above and by Referee #2, my preference is to:

- 1) retain the horizontal stacking of the 3 study groups, but separate the groups via breaks between the x axes of the figure panels,
- 2) retain separation of the two ECC groups, rather than combining them into a single panel, as this is the easiest way to convey the changes between sequential events within each group, which is a major focus of the manuscript,
- 3) retain the column format in each panel, but in the pulmonary arterial blood flow and pulmonary vascular conductance figure panels, highlight non-ventilatory events using shading, and include a note in the figure legend specifying that this shading identifies the non-ventilatory events of the birth transition,
- 4) move the original x-axis labelling into the body of figures, and use clearer descriptive terminology in this labelling,
- 5) place the group names at the top of the figure panels in boxes, and also change ECC_{NA} to “Non-asphyxial ECC” and ECC_A to “Asphyxial ECC”.

4 - Please make clear if food/water was needed/provided as these are acute experiments. Please confirm access while animals were being housed prior to experiments.

In accord with this request, the Methods section of the revised manuscript now states that 1) while being acclimatized for at least 4 days prior to the acute experimental study in individual pens within the Murdoch Children’s Research Institute Translational Research Unit, pregnant

ewes had *ad libitum* access to feed and water, and 2) ewes were fasted overnight prior to the acute study, with *ad libitum* access to water.

Senior Editor

Thank you for submission. Please address all the comments raised by the reviewers, and in particular those related to alternate ways to better display and analyse the data. All reviewers have raised the need to clarify various aspects regarding the data. While these are acute experiments, please provide housing and care information if animals were held in your unit prior to the experiments as per The Journal's guidelines.

The comments and suggestions of the Referees and Reviewing Editor have been addressed, with additional information provided in the revised manuscript on the housing and care of pregnant ewes prior to the acute experimental study.

Referee #1:

Comment: Smolich presents an elegant series of studies which evaluate the contribution of reduction in lung volume, delivery and cord clamping, as well as ventilation, on changes in central hemodynamics, and most notably pulmonary blood flow in an acute preterm lamb surgical preparation. The investigator uses a well-established model for this studies. Data are presented clearly and extend our understanding of hemodynamic changes associate with birth. Thank you for these positive comments about the study.

There are a few points that require clarification.

1. Was maternal administration of glucocorticoids used? For the delivery of most preterm infants, antenatal glucocorticoids, which impact pulmonary function are provided. Though mainly provide to promote surfactant production, this does appear to affect lung water through ENaC.

No, antenatal glucocorticoids (steroids) were not administered to pregnant ewes in this study. Most experimental birth transition studies to date that have included changes in pulmonary blood flow in either lambs e.g. (Crossley *et al.*, 2009a; Sobotka *et al.*, 2011; Bhatt *et al.*, 2013; Smolich *et al.*, 2015; Smolich & Kenna, 2022) or rabbit kittens (Lang *et al.*, 2014; Lang *et al.*, 2016; Lang *et al.*, 2017) have been performed in the absence of antenatal glucocorticoids, largely because this highlights the effect of prematurity on measured variables. The few studies incorporating changes in pulmonary blood flow during the birth transition in lambs that have specifically addressed the effects of antenatal steroids have been performed in the setting of early cord clamping (ECC) with a non-asphyxial interval before ventilation (Crossley *et al.*, 2009b; Smolich *et al.*, 2019) or, more recently, delayed cord clamping (Smolich & Kenna, 2025). There has been no published study to date on the pulmonary blood flow effects of antenatal steroids with ECC and an asphyxial interval before ventilation, so this is an area that warrants attention in a future study.

As antenatal steroids may alter the composition and clearance of lung liquid (Bonanno & Wapner, 2009), increase umbilical blood flow (Schwab *et al.*, 2006) and increase the resting level of pulmonary blood flow in the fetus (Crossley *et al.*, 2009b; Smolich & Mynard, 2021), whether this therapy alters the contribution of non-ventilatory events such as a reduction in lung liquid volume or umbilical cord clamping to rises in pulmonary blood flow with either early or delayed cord clamping during the birth transition is an important question that clearly needs to be addressed in a separate study.

2. The timing of the AoT blood sampling in ECCNA and ECCA are confusing. 20 seconds after clamping in ECCNA with ventilation at 36 sec, and 30 seconds after clamping in ECCA

with ventilation at 100 sec. Why so soon in ECCA relative to ventilation time? Thus, blood gas may not be entirely reflective of PaO₂ at time of ventilation in ECCA group - but explains the relatively small differences in blood gas values between groups (Table 2).

Apologies for the confusion. To clarify the timing of blood gas sampling, in both the non-asphyxial (ECC_{NA}) and asphyxial early cord clamping group (ECC_A), an aortic blood gas sample was withdrawn in the pre-asphyxial phase after ECC. In the ECC_{NA} group, this was at the 20 s time-point, with ventilation started ~15 s later. Because the interval between cord clamping and the start of ventilation was much longer in the ECC_A group, it was possible to delay withdrawal of this initial aortic blood gas sample to 30 s after ECC. This minor difference in timing (i.e. 20 s vs. 30 s after ECC) accounts for O₂ saturation and P_aO₂ at the first timepoint after cord clamping being slightly higher in the ECC_{NA} group than in the ECC_A group in Table 2. The next aortic blood gas sample in the ECC_A group was taken after an asphyxial level of arterial oxygenation had become well-established ~70 s later, just before the start of ventilation, and this is the reason why the O₂ saturation and P_aO₂ at this second time-point after cord clamping were quite low.

3. The ventilatory settings (or at least upper limits thereof) are much higher than would typically be used for preterm infants. I'm curious as to why this was necessary as such high ventilator pressures, particularly in a setting of an open chest, likely impact pulmonary vascular resistance and thus central hemodynamics.

Yes, the pressures employed during mechanical ventilation in our studies were higher than would be used for human infants. However, the use of a relatively high level of PEEP (8 cmH₂O) during mechanical ventilation was physiologically appropriate to optimize dynamic compliance of the lungs in preterm lambs that had not received antenatal steroid treatment at the gestational age used in our studies (Tingay *et al.*, 2014). Furthermore, the maximum peak inspiratory pressure was set at 50 cmH₂O, which is consistent with the approach used in other preterm birth transition studies in lambs, e.g. (Ball *et al.*, 2010; Barton *et al.*, 2014; Tingay *et al.*, 2014). However, this maximal pressure is not attained in studies within our laboratory, as the actual values of peak inspiratory pressure in preterm newborn lambs in the absence of antenatal steroid therapy range between averages of 30-39 cmH₂O (Smolich *et al.*, 2019; Smolich & Kenna, 2025). The levels of PEEP and peak inspiratory pressure were similar in all three groups of the present study, so ventilatory settings are very unlikely to have been responsible for any differences in blood pressures or blood flows present between the groups.

The issues related to ventilatory pressures used in the study have now been clarified within the Methods and Discussion sections of the revised manuscript.

Please also note that, as detailed in a response to a subsequent comment, the experimental preparation used in this study is not “open-chest” in the conventional sense because:

- 1) while the left thoracotomy was not surgically closed, the footprint of this thoracotomy was quite small, overlaying just the apical portion of the left upper lung lobe,
- 2) the right pleural sac was not opened during surgical preparation, so the right lung remained under closed-chest conditions during positive pressure ventilation.

4. The hemodynamic recording periods are relatively short (20-30 seconds, though necessitated by the design of the study). How much intra-animal variability was there in the measurements of various blood flows during this period?

Yes, short recording periods were necessary because of the very rapid changes in haemodynamics and blood flows which occur during the birth transition, particularly with ECC followed by an asphyxial level of arterial oxygenation before the onset of ventilation.

However, these short recording periods were also adequate for steady-state periods before and after the birth transition because, as evident in Fig. 3 of (Smolich *et al.*, 2015), blood pressures and blood flows are quite stable during these periods in our experimental preparation.

In response to the Referee's query about intra-animal variability of blood flows, beat-by-beat measurements in the various stages of the birth transition have been performed on right ventricular output and peak, mean and pulse (i.e. maximum minus minimum) pulmonary arterial blood flows in animals of the three study groups, using the same data epochs that were utilized for generation of ensemble-averaged signals. As evident in Tables 1-3 below, intra-animal variability was relatively minor, with coefficients of variation (CoV) in most instances of $\leq 5\%$. The exception was mean pulmonary arterial blood flow, where the CoV was higher (28-48%) in the baseline fetal state (and also during asphyxia), but progressively lower after reduction of lung liquid volume and fetal delivery. However, the CoV of corresponding peak and pulse mean pulmonary arterial blood flows was $< 5\%$, suggesting that higher values for mean pulmonary arterial blood flows were related to large beat-to-beat variation in pulmonary arterial backflow, which resides within the body of the pulmonary arterial blood flow profile.

Table 1. Coefficient of variation (CoV) of pulmonary arterial (PA) blood flows and right ventricular (RV) output in non-ventilatory and ventilatory phases of birth transition with non-asphyxial early cord clamping.

Variable	CoV peak PA flow	CoV pulse PA flow	CoV mean PA flow	CoV RV output
Baseline	2(1)%	1(1)%	28(19)%	2(2)%
↓Lung liquid volume	1(1)%	2(1)%	16(13)%	3(2)%
Post-delivery	1(1)%	2(2)%	8(5)%	2(2)%
CC1	2(1)%	3(2)%	5(4)%	2(1)%
CC2	2(1)%	4(3)%	7(6)%	4(4)%
Post ventilation	4(2)%	5(2)%	7(3)%	4(1)%
Peak newborn PA flow	6(2)%	7(3)%	3(1)%	7(3)%

Data are expressed as means(SD); n = 10. Abbreviations: CC1, within initial 30 s after early cord clamping; CC2, just before ventilation after early cord clamping.

Table 2. Coefficient of variation (CoV) of pulmonary arterial (PA) blood flows and right ventricular (RV) output in non-ventilatory and ventilatory phases of birth transition with asphyxial early cord clamping .

Variable	CoV peak PA flow	CoV pulse PA flow	CoV mean PA flow	CoV RV output
Baseline	2(1)%	2(1)%	48(20)%	2(1)%
↓Lung liquid volume	2(2)%	2(2)%	23(12)%	2(1)%
Post-delivery	2(1)%	3(2)%	10(7)%	2(1)%
CC1	3(2)%	5(4)%	8(7)%	2(1)%
CC2	3(2)%	5(4)%	28(33)%	3(2)%
Post ventilation	7(4)%	8(3)%	9(4)%	5(2)%
Peak newborn PA flow	6(2)%	8(3)%	4(2)%	6(2)%

Data are expressed as means(SD); n = 10. Abbreviations: CC1, within initial 30 s after early cord clamping; CC2, just before ventilation after early cord clamping.

Table 3. Coefficient of variation (CoV) of pulmonary arterial (PA) blood flows and right ventricular (RV) output in non-ventilatory and ventilatory phases of birth transition with delayed cord clamping .

Variable	CoV peak PA flow	CoV pulse PA flow	CoV mean PA flow	CoV RV output
Baseline	1(1)%	1(1)%	34(20)%	2(1)%
↓Lung liquid volume	1(1)%	2(1)%	16(10)%	2(1)%
Post-delivery	1(1)%	2(1)%	8(4)%	2(1)%
V1	2(1)%	3(1)%	5(3)%	2(1)%
V2	2(1)%	3(1)%	6(2)%	3(2)%
Post cord clamping	3(1)%	5(1)%	4(1)%	5(2)%
Peak newborn PA flow	4(1)%	5(1)%	3(1)%	6(2)%

Data are expressed as means(SD); n = 11. Abbreviations: V1, within first 30 s after onset of initial ventilation; V2, during initial ventilation, just before delayed cord clamping.

5. Were there animal preparations for which data were not included? If so, how many and why?

Animals included in the present study were randomly selected from a pool of relevant early and delayed cord clamping birth transition experiments performed in the period spanning 2015 to 2021, with data then specifically analyzed for this study. *A priori* criteria for inclusion of experimental studies within this data analysis were that full fetal instrumentation with the required pressure catheters and flow probes was able to be performed, that fetal aortic blood gas variables after completion of surgery were within widely-accepted normal ranges (viz. pH \geq 7.3, Pao₂ \geq 20 mmHg, Paco₂ \leq 55 mmHg), that high-quality data recordings were available for all non-ventilatory events during the birth transition, including before and after reduction of lung liquid volume, and that animals underwent an uneventful and uncomplicated cesarean section birth delivery. A target animal number of 10-11 per study group was chosen on the basis that this cohort size permitted robust statistical analysis of blood flow data of interest, as evident in other recent fetal and birth transition studies from this laboratory (Smolich & Mynard, 2019; Smolich *et al.*, 2021; Smolich & Kenna, 2022; Smolich *et al.*, 2024).

6. Consider softening the comment regarding the data refuting the view that an increase in pulmonary perfusion.....lung aeration (lines 498-99). In reviewing these references, though the authors conclude ventilation is the primary stimulus for increase in blood flow, I don't believe they comment about an absence of increase in pulmonary blood flow until aeration. They also demonstrate that aeration is not needed to increase blood flow as they demonstrate, in their model, that blood flow increases in areas of the lung not aerated (or expanded). I recognize Smolich studied an immediate period of time after delivery (and before) that Lang and Hooper did not.

The sentence referred to in lines 498-499 reads “The results of the current study do not support the view that an increase in pulmonary perfusion at birth is not triggered until the onset of lung aeration (Hooper *et al.*, 2015a; Hooper *et al.*, 2015b; Hooper *et al.*, 2015c)”.

The basis of this sentence is that the cited reviews (Hooper *et al.*, 2015a; Hooper *et al.*, 2015b; Hooper *et al.*, 2015c) all emphasize the very low level of pulmonary blood flow present in the fetus, and then include statements such as:

- 1) “As it also triggers the increase in pulmonary blood flow (PBF) at birth, lung aeration is the central determining event for the successful transition to newborn life” and “...it is well established that lung aeration triggers the increase in PBF after birth...” (Hooper *et al.*, 2015a),
- 2) “..lung aeration...triggers a huge decrease in PVR and an increase in PBF” (Hooper *et al.*, 2015b),
- 3) “Lung aeration triggers the decrease in PVR and increase in PBF” at birth, “The rationale underpinning all of the proposed mechanisms responsible for the increase in PBF at birth suggests that the response is solely activated by air entry...” and “As the increase in PBF (at birth) results from lung aeration...” (Hooper *et al.*, 2015c).

Furthermore, there is no mention in these reviews of any other factor that might contribute to an increase of pulmonary blood flow during the birth transition. Thus, although these reviews do not specifically state that there is an absence of an increase in pulmonary blood flow until aeration, it is very difficult to see how a reader (particularly one not familiar with the area) could draw any conclusion other than that an increase in pulmonary blood flow at birth was not triggered until the onset of lung aeration.

It is also important to point out that the remainder of the paragraph starting with the sentence in lines 498-499 seeks to explain why the experimental design employed in the studies of Lang *et al.* (Lang *et al.*, 2016; Lang *et al.*, 2017), on which the reviews of Hooper *et al.* (Hooper *et al.*, 2015a; Hooper *et al.*, 2015b; Hooper *et al.*, 2015c) are primarily based, would lead to the conclusion that it is solely lung aeration which triggers the increase in pulmonary blood flow at birth.

Finally, while Lang *et al.* (2014) did demonstrate that increases in pulmonary blood flow can occur without aeration in rabbit kittens, this was in the setting of a very non-physiological situation that is not part of a normal birth transition, as this increase in pulmonary blood flow was observed in the non-aerated lung during selective ventilation of the opposite lung. Moreover, the presence of a profound bradycardia (average heart rate of 69 beats/min) in rabbit kittens of Lang *et al.* (2014) before ventilation indicated that this ventilation occurred in the presence of a pronounced asphyxial state. Thus, it is quite possible that an increase in pulmonary blood flow observed within the non-ventilated lung was simply a consequence of increased arterial oxygenation arising from selective ventilation of the other lung.

Given the above points, I would contend that the sentence in lines 498-499 is accurate, fair and reasonable, and therefore warrants retention in the revised manuscript.

7. The author appropriately acknowledges limitations of his model. In listing these (lines 520-22), please include open-chested and mechanically ventilated with positive pressure for completeness. It should also be restated that the experiments were performed in preterm animals in the absence of the effect of labor and natural decrease in lung liquid volume that occurs as the fetus approaches term.

Information on additional limitations of the study has been provided in the Discussion, with the emphasis that study findings relate to preterm birth.

Please note that, rather than use of the term “open-chested”, it has been stated that the thoracotomy was not surgically closed. This is because “open-chested” is not an accurate

description of the experimental preparation, as the area where the chest was actually open in the third left interspace overlay just the apical portion of the left lung, which constitutes ~20% of the left lung at most. Furthermore, the right pleural cavity was not opened during surgical preparation, so that the right lung, which is about 50% larger than the left lung and therefore constitutes ~60% of total lung weight, was subjected to closed-chest conditions throughout the period of positive-pressure ventilation.

8. I appreciate the clinical implication comment regarding earlier vs later initiation of ventilation to enhance PBF. What may be of interest, and perhaps unknown (though not likely) to the investigator, is that two large trials of early ventilation/resuscitation in preterm infants failed to have an effect of study outcomes (though PBF was obviously not assessed, including intraventricular hemorrhage. (PMID 39671198, 38758557). Given the data regarding improvements in hemodynamic fluctuations shown in the present study and others by the investigator by early ventilation, these findings were very disappointing.

The frustration expressed by the Referee as regards beneficial effects of ventilation before cord clamping suggested by experimental studies not being translated into improved neonatal outcomes (including incidence of intraventricular haemorrhage) is understandable. However, at least two factors are likely to be implicated in this situation:

- 1) Clinical adverse events generally occur hours to days after birth, whereas experimental studies are primarily concerned with changes and differences in specific physiological variables that are measured over a much shorter timeframe. Disturbances in physiological variables during the birth transition are thus likely to be just one component of a multifactorial cascade contributing to emergence of an adverse event or an alteration in neonatal outcome. It is therefore perhaps not surprising that the clinical studies of Fairchild *et al.* (2024) and Pratesi *et al.* (2024) have not shown a clear benefit of initial ventilation before DCC on neonatal outcome or the incidence of intraventricular haemorrhage.
- 2) With potentially a large degree of data variability in multi-centre trials related to factors such as differences in clinical practice, infant demographics, levels of monitoring etc, a very large patient cohort is likely to be a requirement for detecting statistically significant changes in neonatal outcome. Thus, the meta-analysis of Seidler *et al.* (2023), which concluded that a delay of 120-180 s before DCC did reduce the incidence of death before hospital discharge compared to early cord clamping, cord milking or shorter periods of delay in preterm neonates, utilized data from 47 trials that involved >6000 participants. By contrast, 570 infants were studied in Fairchild *et al.* (2024) and 209 infants in Pratesi *et al.* (2024).

Referee #2:

This study used acutely-instrumented preterm lambs to quantify the contribution of a reduction in lung liquid volume, fetal delivery and cord clamping, as well as ventilation to the increase in pulmonary blood flow at birth.

These interesting results confirm this respondent's bias that ventilation is the major factor, but provide important and novel insight by quantifying a roughly 30% contribution from multiple other factors.

This complex study is performed appropriately, using well validated methods.

The major limitation is the use of anaesthesia, which abolishes reflex pathways. The author acknowledges and carefully discusses this important issue.

Thanks are extended to the Referee for these positive comments about the study. As acknowledged in the manuscript, ventilation/aeration is clearly the major factor contributing to a large rise in pulmonary blood flow occurring at birth. However, contrary to the messaging contained in cited reviews (Hooper *et al.*, 2015a; Hooper *et al.*, 2015b; Hooper *et al.*, 2015c), the present study suggests that it is not the sole factor involved in this rise, and for the first time, quantifies the contribution of non-ventilatory events to this rise.

Ibid. Please include 'anaesthetised' in the title and abstract.

Unfortunately, it is not possible to include 'anaesthetised' in the title, as this already comprises the maximal permissible 150 characters and spaces, and there is no wording which can easily be removed without compromising the fundamental message of the title. However, 'anaesthetised' has now been included in the abstract of the revised manuscript. It should be noted that use of anaesthesia in these types of studies is not confined to this laboratory, as birth transition and neonatal studies performed in large laboratory animals with positive-pressure mechanical ventilation require anaesthesia, e.g. (Crossley *et al.*, 2009a; Bhatt *et al.*, 2013; Eiby *et al.*, 2023).

It isn't clear to this non-statistician how the contributions of each factor were quantified. I think it is based on crude means at different times. It has been suggested that a mixed model analysis can be more robust in this setting? I respectfully suggest that obtaining professional statistical input may be helpful for such a complex, multifactorial analysis of longitudinal data, and that a more refined analysis can only improve the impact of this very interesting ms.

I presume that by "crude means", the Referee considers that the means for each non-ventilatory event were used to calculate the contribution of these events to the increase in pulmonary blood flow between the baseline fetal level and the initial neonatal peak. This is not the case. Instead, the increments related to non-ventilatory events were calculated in each individual animal, summed in each individual animal, and then related to the rise in pulmonary blood flow between the baseline fetal state and the peak neonatal value in each animal (subtraction of this percentage from 100 yielded the contribution of ventilation to the increase in PA flow within each individual animal). The percentages shown in Fig 7 thus represent the mean and SD of the ventilatory and non-ventilatory contributions obtained using individual animal data in each group. This has now been clarified in the Methods section of the revised manuscript.

Statistical advice has also been sought on whether mixed modelling analysis, rather than one-way repeated measures analysis of variance used in the original submission, might be a better approach to analyze data in this study. In this context, it is important to remember that the present study was focused on changes in haemodynamics and blood flows occurring at specific time-points within each of the three study groups between the baseline fetal state and the initial peak in pulmonary arterial blood flow occurring 10-15 minutes after birth. Because of this focus, detailed statistical analysis of differences in changes between study groups was intentionally avoided for two main reasons:

- 1) These differences have been addressed in prior studies (cited in the current manuscript) that compared non-asphyxial early cord clamping with asphyxial early cord clamping (Smolich *et al.*, 2015), and non-asphyxial early cord clamping with delayed cord clamping (Smolich & Kenna, 2022). Note that 1) the pattern of pulmonary arterial and ductal blood flows in these birth transition studies (Smolich *et al.*, 2015; Smolich & Kenna, 2022) were

very similar to those observed in the present study, and 2) a study comparing changes in haemodynamics and blood flows between asphyxial early cord clamping and delayed cord clamping has not been performed, as this comparison is not very meaningful physiologically or clinically, given the marked differences in blood flow responses that exist between non-asphyxial and asphyxial early cord clamping (Smolich *et al.*, 2015; Smolich *et al.*, 2020).

- 2) Detailed comparison of changes between groups would not only considerably increase the length of the Results and Discussion sections, but also detract from a main message of the manuscript, which is that non-ventilatory events make a substantial (and similar) contribution to increases in pulmonary blood flow during the birth transition *irrespective* of the birth strategy employed.

In respect of the choice of statistical approach, with complete datasets (which was the case in the present study), mixed modelling analysis provides the same overall statistical outcome as one-way repeated measures analysis of variance (as demonstrated in the online GraphPad Prism statistics guide). However, determining the presence of statistical differences between specific time-points in each group within this overall outcome, particularly if there is no need for evaluation of inter-individual differences, is much more straightforward with one-way repeated measures analysis of variance. Thus, based on the “rule of simplicity” (i.e. with multiple possible approaches, the simplest one is likely to be the best), one-way repeated measures analysis of variance has been preferred in the manuscript.

Minor.

Figures; please replace bar graphs with vertical scatter plots, with median or mean bars.

As acknowledged in the response to comments of the Reviewing Editor, the figure format of the original submission did not easily and adequately convey the main messages of the study. After considering the constructive suggestions of the Reviewing Editor and the Referee, my preference is to retain the presentation of data in columns, but with demarcation of non-ventilatory components via shading, and with repositioning of column labelling.

Unfortunately, vertical scatter plots suggested by Referee #2 are not visually appealing with the data of this manuscript, because for many variables (e.g. fetal baseline pulmonary blood flow and pulmonary vascular conductance in Fig. 3, left-to-right ductal flow before cord clamping in the early cord clamping groups or before ventilation in the delayed cord clamp group in Fig. 6), values are very close to the x-axis. Datapoints large enough to be easily seen in a scatter plot thus straddle the x-axis.

Trivia.

there are too many nonstandard abbreviations that make the MS hard to follow. DCC means something different to this respondent for example. ECC and DCC in particular should be spelt out please, as should nearly all the remainder.

Yes, I readily acknowledge that the abbreviations ECC and DCC both have multiple meanings within the medical literature. However, in the birth transition physiology and neonatology fields, the use of ECC as an abbreviation for “early cord clamping” and DCC as an abbreviation for “delayed cord clamping” is widely accepted and has been routinely employed in publications within many journals e.g. (Pichler *et al.*, 2015; Blank *et al.*, 2018; Katheria *et al.*, 2018; Rana *et al.*, 2018; Bates *et al.*, 2019; Katheria *et al.*, 2019; Kc *et al.*, 2019; Lodha *et al.*, 2019; Nudelman *et al.*, 2019; Rabe *et al.*, 2019; Rana *et al.*, 2019; Tang *et al.*, 2019; El-Naggar *et al.*, 2020; Giovannini *et al.*, 2020; Badurdeen *et al.*, 2021; Gomersall

et al., 2021; Jasani *et al.*, 2021; Badurdeen *et al.*, 2022; Knol *et al.*, 2022; Chaudhary *et al.*, 2023; Fairchild, 2023; Garcia *et al.*, 2023; Badurdeen *et al.*, 2024). My preference is therefore to retain these abbreviations, particularly as replacing them with the full wording would increase the length of the manuscript by at least half a page.

In view of the Referee's comment, however, "ECC_{NA}" has been replaced with "non-asphyxial ECC" and "ECC_A" with "asphyxial ECC" throughout the revised manuscript, while other non-standard abbreviations have been replaced with their full wording. Thus, the main abbreviations retained in the revised manuscript are ECC, DCC, PA (for 'pulmonary arterial') and RV (for 'right ventricular').

"with most of this output" unclear. What output? Best guess that you mean of total RV output here

Yes, this phrase in the Introduction does refer to RV output, as it is the only output mentioned in the sentence. However, to avoid any potential confusion, the phrase has been changed to read "with most RV output".

"complete delivery of the fetus from the uterus, which augments pulmonary arterial (PA) blood flow and RV output," how is this likely to work? Suggest briefly suggest the likely mechanism here, to lead in to the detailed discussion.

The reference cited in the Introduction section for the statement contained within this comment (Smolich & Kenna, 2022) reported these changes in PA flow and RV output, but did not address their potential mechanism. I would therefore prefer not to suggest a mechanism in the Introduction, as it is not until the Discussion section that a likely mechanism (with supporting references) is proposed.

"ensemble-averaged". What does this mean specifically in this context? There is no discussion of statistical models to be averaged.

To clarify, ensemble-averaging is a standard and powerful signal processing technique which involves datapoint-by-datapoint averaging of a regularly repeating signal (e.g. a blood pressure or blood flow profile within a full cardiac cycle). This technique reduces noise in the signal, thereby increasing the signal-to-noise ratio and thus the signal fidelity (Semmlow, 2018). For each signal that is ensemble-averaged in individual animals within a given time-point or state, this technique produces a single waveform from which the average, maximum, minimum etc, can be determined, with these individual values within groups then used 1) to generate the group mean and SD for time-points or states of interest, and 2) in statistical analyses.

The author may wish to review some emotive language in the results... "then surged".

This phrase has been changed to "then increased" in the arterial blood pressures and heart rate portion of the Results section of the revised manuscript.

References

- Badurdeen S, Blank DA, Hoq M, Wong FY, Roberts CT, Hooper SB, Polglase GR & Davis PG. (2024). Blood pressure and cerebral oxygenation with physiologically-based cord clamping: sub-study of the BabyDUCC trial. *Pediatr Res* **96**, 124-131.
- Badurdeen S, Davis PG, Hooper SB, Donath S, Santomartino GA, Heng A, Zannino D, Hoq M, Omar FKC, Kane SC, Woodward A, Roberts CT, Polglase GR, Blank DA & group BDUCCc. (2022). Physiologically based cord clamping for infants $\geq 32+0$ weeks

- gestation: A randomised clinical trial and reference percentiles for heart rate and oxygen saturation for infants $\geq 35+0$ weeks gestation. *PLoS Med* **19**, e1004029.
- Badurdeen S, Santomartino GA, Thio M, Heng A, Woodward A, Polglase GR, Hooper SB, Blank DA & Davis PG. (2021). Respiratory support after delayed cord clamping: a prospective cohort study of at-risk births at $\geq 35(+0)$ weeks gestation. *Arch Dis Child Fetal Neonatal Ed* **106**, 627-634.
- Ball MK, Hillman NH, Kallapur SG, Polglase GR, Jobe AH & Pillow JJ. (2010). Body temperature effects on lung injury in ventilated preterm lambs. *Resuscitation* **81**, 749-754.
- Barton SK, Moss TJ, Hooper SB, Crossley KJ, Gill AW, Kluckow M, Zahra V, Wong FY, Pichler G, Galinsky R, Miller SL, Tolcos M & Polglase GR. (2014). Protective ventilation of preterm lambs exposed to acute chorioamnionitis does not reduce ventilation-induced lung or brain injury. *PLoS One* **9**, e112402.
- Bates SE, Isaac TCW, Marion RL, Norman V, Gumley JS & Sullivan CD. (2019). Delayed cord clamping with stabilisation at all preterm births - feasibility and efficacy of a low cost technique. *Eur J Obstet Gynecol Reprod Biol* **236**, 109-115.
- Bhatt S, Allison BJ, Wallace EM, Crossley KJ, Gill AW, Kluckow M, te Pas AB, Morley CJ, Polglase GR & Hooper SB. (2013). Delaying cord clamping until ventilation onset improves cardiovascular function at birth in preterm lambs. *J Physiol* **591**, 2113-2126.
- Blank DA, Badurdeen S, Omar FKC, Jacobs SE, Thio M, Dawson JA, Kane SC, Dennis AT, Polglase GR, Hooper SB & Davis PG. (2018). Baby-directed umbilical cord clamping: A feasibility study. *Resuscitation* **131**, 1-7.
- Bonanno C & Wapner RJ. (2009). Antenatal corticosteroid treatment: what's happened since Drs Liggins and Howie? *Am J Obstet Gynecol* **200**, 448-457.
- Chaudhary P, Priyadarshi M, Singh P, Chaurasia S, Chaturvedi J & Basu S. (2023). Effects of delayed cord clamping at different time intervals in late preterm and term neonates: a randomized controlled trial. *Eur J Pediatr* **182**, 3701-3711.
- Crossley KJ, Allison BJ, Polglase GR, Morley CJ, Davis PG & Hooper SB. (2009a). Dynamic changes in the direction of blood flow through the ductus arteriosus at birth. *J Physiol* **587**, 4695-4704.
- Crossley KJ, Morley CJ, Allison BJ, Davis PG, Polglase GR, Wallace MJ, Zahra VA & Hooper SB. (2009b). Antenatal corticosteroids increase fetal, but not postnatal, pulmonary blood flow in sheep. *Pediatr Res* **66**, 283-288.
- Eiby YA, Wright IMR, Stark MJ & Lingwood BE. (2023). Red cell infusion but not saline is effective for volume expansion in preterm piglets. *Pediatr Res* **94**, 112-118.
- El-Naggar W, Afifi J, Dorling J, Bodani J, Cieslak Z, Canning R, Ye XY, Crane J, Lee SK, Shah PS, Canadian Neonatal N & the Canadian Preterm Birth Network I. (2020). A Comparison of Strategies for Managing the Umbilical Cord at Birth in Preterm Infants. *J Pediatr* **225**, 58-64 e54.
- Fairchild K. (2023). Assisted ventilation prior to umbilical cord clamping: Potential benefits, challenges, and research studies. *Semin Perinatol* **47**, 151788.
- Fairchild KD, Petroni GR, Varhegyi NE, Strand ML, Josephsen JB, Niermeyer S, Barry JS, Warren JB, Rincon M, Fang JL, Thomas SP, Travers CP, Kane AF, Carlo WA, Byrne BJ, Underwood MA, Poulain FR, Law BH, Gorman TE, Leone TA, Bulas DI, Epelman M, Kline-Fath BM, Chisholm CA, Kattwinkel J & VentFirst C. (2024). Ventilatory assistance

- before umbilical cord clamping in extremely preterm infants: a randomized clinical trial. *JAMA Netw Open* **7**, e2411140.
- Garcia C, Prieto MT, Escudero F, Bosh-Gimenez V, Quesada L, Lewanczyk M, Pertegal M, Delgado JL, Blanco-Carnero JE & De Paco Matallana C. (2023). The impact of early versus delayed cord clamping on hematological and cardiovascular changes in preterm newborns between 24 and 34 weeks' gestation: a randomized clinical trial. *Arch Gynecol Obstet*.
- Giovannini N, Crippa BL, Denaro E, Raffaelli G, Cortesi V, Consonni D, Cetera GE, Parazzini F, Ferrazzi E, Mosca F & Ghirardello S. (2020). The effect of delayed umbilical cord clamping on cord blood gas analysis in vaginal and caesarean-delivered term newborns without fetal distress: a prospective observational study. *BJOG* **127**, 405-413.
- Gomersall J, Berber S, Middleton P, McDonald SJ, Niermeyer S, El-Naggar W, Davis PG, Schmolzer GM, Ovelman C, Soll RF & International Liaison Committee On Resuscitation Neonatal Life Support Task F. (2021). Umbilical Cord Management at Term and Late Preterm Birth: A Meta-analysis. *Pediatrics* **147**, e2020015404.
- Hooper SB, Polglase GR & Roehr CC. (2015a). Cardiopulmonary changes with aeration of the newborn lung. *Paediatr Respir Rev* **16**, 147-150.
- Hooper SB, Polglase GR & te Pas AB. (2015b). A physiological approach to the timing of umbilical cord clamping at birth. *Arch Dis Child Fetal Neonatal Ed* **100**, F355-360.
- Hooper SB, Te Pas AB, Lang J, van Vonderen JJ, Roehr CC, Kluckow M, Gill AW, Wallace EM & Polglase GR. (2015c). Cardiovascular transition at birth: a physiological sequence. *Pediatr Res* **77**, 608-614.
- Jasani B, Torgalkar R, Ye XY, Syed S & Shah PS. (2021). Association of umbilical cord management strategies with outcomes of preterm infants: a systematic review and network meta-analysis. *JAMA Pediatr* **175**, e210102.
- Katheria A, Hosono S & El-Naggar W. (2018). A new wrinkle: Umbilical cord management (how, when, who). *Semin Fetal Neonatal Med* **23**, 321-326.
- Katheria AC, Rich WD, Bava S & Lakshminrusimha S. (2019). Placental transfusion for asphyxiated infants. *Front Pediatr* **7**, 473.
- Kc A, Singhal N, Gautam J, Rana N & Andersson O. (2019). Effect of early versus delayed cord clamping in neonate on heart rate, breathing and oxygen saturation during first 10 minutes of birth - randomized clinical trial. *Matern Health Neonatol Perinatol* **5**, 7.
- Knol R, Brouwer E, van den Akker T, DeKoninck PLJ, Lopriore E, Onland W, Vermeulen MJ, van den Akker-van Marle ME, van Bodegom-Vos L, de Boode WP, van Kaam AH, Reiss IKM, Polglase GR, Hutten GJ, Prins SA, Mulder EEM, Hulzebos CV, van Sambeek SJ, van der Putten ME, Zonnenberg IA, Hooper SB & Te Pas AB. (2022). Physiological-based cord clamping in very preterm infants: the Aeration, Breathing, Clamping 3 (ABC3) trial-study protocol for a
- Lang JA, Pearson JT, Binder-Heschl C, Wallace MJ, Siew ML, Kitchen MJ, Te Pas AB, Fouras A, Lewis RA, Polglase GR, Shirai M & Hooper SB. (2016). Increase in pulmonary blood flow at birth; role of oxygen and lung aeration. *J Physiol* **594**, 1389-1398.
- Lang JA, Pearson JT, Binder-Heschl C, Wallace MJ, Siew ML, Kitchen MJ, Te Pas AB, Lewis RA, Polglase GR, Shirai M & Hooper SB. (2017). Vagal denervation inhibits the

- increase in pulmonary blood flow during partial lung aeration at birth. *J Physiol* **595**, 1593-1606.
- Lang JA, Pearson JT, te Pas AB, Wallace MJ, Siew ML, Kitchen MJ, Fouras A, Lewis RA, Wheeler KI, Polglase GR, Shirai M, Sonobe T & Hooper SB. (2014). Ventilation/perfusion mismatch during lung aeration at birth. *J Appl Physiol* **117**, 535-543.
- Lodha A, Shah PS, Soraisham AS, Rabi Y, Abou Mehrem A, Singhal N & Canadian Neonatal Network I. (2019). Association of deferred vs immediate cord clamping with severe neurological injury and survival in extremely low-gestational-age neonates. *JAMA Netw Open* **2**, e191286.
- Nudelman MJ, Goel K, Jegatheesan P, Song D, Huang A & Govindaswami B. (2019). Haematocrit in <35 weeks preterm infants who received at least 60 seconds of delayed cord clamping: a retrospective observational study. *BMJ Paediatr Open* **3**, e000531.
- Pichler G, Baik N, Urlesberger B, Cheung PY, Aziz K, Avian A & Schmolzer GM. (2015). Cord clamping time in spontaneously breathing preterm neonates in the first minutes after birth: impact on cerebral oxygenation - a prospective observational study. *J Matern Fetal Neonatal Med*, 1-3.
- Pratesi S, Ciarcia M, Boni L, Ghirardello S, Germini C, Troiani S, Tulli E, Natile M, Ancora G, Barone G, Vedovato S, Bertuola F, Parata F, Mescoli G, Sandri F, Corbetta R, Ventura L, Dognini G, Petrillo F, Valenzano L, Manzari R, Lavizzari A, Mosca F, Corsini I, Poggi C, Dani C & Collaborators PCIT. (2024). Resuscitation with placental circulation intact compared with cord milking: a randomized clinical trial. *JAMA Netw Open* **7**, e2450476.
- Rabe H, Gyte GM, Diaz-Rossello JL & Duley L. (2019). Effect of timing of umbilical cord clamping and other strategies to influence placental transfusion at preterm birth on maternal and infant outcomes. *Cochrane Database Syst Rev* **9**, CD003248.
- Rana A, Agarwal K, Ramji S, Gandhi G & Sahu L. (2018). Safety of delayed umbilical cord clamping in preterm neonates of less than 34 weeks of gestation: a randomized controlled trial. *Obstet Gynecol Sci* **61**, 655-661.
- Rana N, Kc A, Malqvist M, Subedi K & Andersson O. (2019). Effect of Delayed Cord Clamping of Term Babies on Neurodevelopment at 12 Months: A Randomized Controlled Trial. *Neonatology* **115**, 36-42.
- Schwab M, Coksaygan T & Nathanielsz PW. (2006). Betamethasone effects on ovine uterine and umbilical placental perfusion at the dose used to enhance fetal lung maturation. *Am J Obstet Gynecol* **194**, 572-579.
- Seidler AL, Libesman S, Hunter KE, Barba A, Aberoumand M, Williams JG, Shrestha N, Aagerup J, Sotiropoulos JX, Montgomery AA, Gyte GML, Duley L, Askie LM & i CC. (2023). Short, medium, and long deferral of umbilical cord clamping compared with umbilical cord milking and immediate clamping at preterm birth: a systematic review and network meta-analysis with individual participant data. *Lancet* **402**, 2223-2234.
- Semmlow J. (2018). Signal analysis in the time domain. In *Circuits, signals and systems for bioengineers*, 3rd edn, pp. 51-106. Academic Press, London.
- Smolich JJ, Cheung MMH & Mynard JP. (2021). Reducing lung liquid volume in fetal lambs decreases ventricular constraint. *Pediatr Res* **90**, 795-800.

- Smolich JJ & Kenna KR. (2022). Divergent effects of initial ventilation with delayed cord clamping on systemic and pulmonary arterial flows in the birth transition of preterm lambs. *J Physiol* **600**, 3583-3601.
- Smolich JJ & Kenna KR. (2025). Antenatal betamethasone augments lung perfusion but lowers upper body blood flow and O₂ delivery with delayed cord clamping at birth in preterm lambs. *J Physiol* **603**, 949-970.
- Smolich JJ, Kenna KR & Cheung MM. (2015). Onset of asphyxial state in non-respiring interval between cord clamping and ventilation increases hemodynamic lability of birth transition in preterm lambs. *J Appl Physiol (1985)* **118**, 675-683.
- Smolich JJ, Kenna KR, Cheung MMH & Mynard JP. (2020). Brief asphyxial state following immediate cord clamping accelerates onset of left-to-right shunting across the ductus arteriosus after birth in preterm lambs. *J Appl Physiol (1985)* **128**, 429-439.
- Smolich JJ, Kenna KR & Mynard JP. (2019). Antenatal betamethasone augments early rise in pulmonary perfusion at birth in preterm lambs: role of ductal shunting and right ventricular outflow distribution. *Am J Physiol Regul Integr Comp Physiol* **316**, R716-R724.
- Smolich JJ, Kenna KR & Mynard JP. (2024). Extended period of ventilation before delayed cord clamping augments left-to-right shunting and decreases systemic perfusion at birth in preterm lambs. *J Physiol* **602**, 1791-1813.
- Smolich JJ & Mynard JP. (2019). Reducing lung liquid volume increases biventricular outputs and systemic arterial blood flows despite decreased cardiac filling pressures in fetal lambs. *Am J Physiol Regul Integr Comp Physiol* **316**, R274-R280.
- Smolich JJ & Mynard JP. (2021). Antenatal betamethasone redistributes central blood flows and preferentially augments right ventricular output and pump function in preterm fetal lambs. *Am J Physiol Regul Integr Comp Physiol* **320**, R611-618.
- Sobotka KS, Hooper SB, Allison BJ, Te Pas AB, Davis PG, Morley CJ & Moss TJ. (2011). An initial sustained inflation improves the respiratory and cardiovascular transition at birth in preterm lambs. *Pediatr Res* **70**, 56-60.
- Tang J, Fullarton R, Samson SL & Chen Y. (2019). Delayed cord clamping does not affect umbilical cord blood gas analysis. *Arch Gynecol Obstet* **299**, 719-724.
- Tingay DG, Bhatia R, Schmolzer GM, Wallace MJ, Zahra VA & Davis PG. (2014). Effect of sustained inflation versus step-wise PEEP strategy at birth on gas exchange and lung mechanics in preterm lambs. *Pediatr Res* **75**, 288-294.

Dear Dr Smolich,

Re: JP-RP-2025-288749R1 "Non-ventilatory events substantially contribute to rise in pulmonary arterial blood flow with early or delayed cord clamping at birth in preterm lambs" by Joseph John Smolich

Thank you for submitting your manuscript to The Journal of Physiology. It has been assessed by a Reviewing Editor and by 2 expert referees and we are pleased to tell you that it is acceptable for publication following satisfactory revision.

REVISION CHECKLIST:

We look forward to receiving your revised submission.

Yours sincerely,

Laura Bennet
Senior Editor
The Journal of Physiology

EDITOR COMMENTS

Reviewing Editor:

Thank you for your careful revisions. The improvements made to the graphing are well appreciated. Congratulations on an excellent study and thank you for your contribution to the Journal.

Senior Editor:

Thank you for your submission. Major amendments have been made. Reviewers highlight one concern about the graphs, but if you feel that the suggested scatterplots would make it harder to read, then we can accept the current graphs. However, you may wish to give it some consideration.

REFEREE COMMENTS

Referee #1:

None

Referee #2:

The author has addressed most of the reviewers' questions in a reasonable way.

Abbreviations. I respectfully disagree that non-standard abbreviations are reasonable to save space. ON behalf of all readers from outside this very narrow field, I beg that author and Editors consider spelling them out. if space needs to be saved, judicious editing would readily save space.

Graphs. The majority of figures are vertical bars. Yes, some of the means are very close to zero. But, of course this will be obvious regardless of the method of presentation, and realistically most are not close to the axis. After careful consideration, I submit that superimposed vertical scatterplots showing the actual data would give the reader much more information. This should be a discussion with the Editors.

END OF COMMENTS

Reviewing Editor

Thank you for your careful revisions. The improvements made to the graphing are well appreciated. Congratulations on an excellent study and thank you for your contribution to the Journal.

Thank you for these comments and the previous suggestion to re-work manuscript figures.

Senior Editor

Thank you for your submission. Major amendments have been made. Reviewers highlight one concern about the graphs, but if you feel that the suggested scatterplots would make it harder to read, then we can accept the current graphs. However, you may wish to give it some consideration.

Thank you for these comments. As mentioned previously, the figures of this manuscript have been quite difficult to display in a form which easily conveys the main messages of the data over the large changes in a number of key variables during the birth transition. Against this background, the column graphs submitted in the last revision were the best overall compromise of the many versions that were examined in the course of that revision. In light of the comments of Referee #2 below, however, the suggested vertical scatterplot presentation has been re-evaluated. As detailed in the response to Referee #2, this approach has two main problems that particularly affect clear depiction of changes in the early, non-ventilatory phases of the birth transition. As these changes (especially pulmonary arterial blood flow and pulmonary vascular conductance) are a main focus of the manuscript, I would therefore prefer that the figures submitted in the last revision of the manuscript be retained.

Referee #1:

Comment: None.

Thank you for accepting the responses to the previous comments about the manuscript.

Referee #2:

The author has addressed most of the reviewers' questions in a reasonable way.

Thank you for this positive assessment of previous responses to comments about the manuscript.

Abbreviations. I respectfully disagree that non-standard abbreviations are reasonable to save space. ON behalf of all readers from outside this very narrow field, I beg that author and Editors consider spelling them out. if space needs to be saved, judicious editing would readily save space.

It was readily acknowledged in the previous response that DCC and ECC (abbreviations for delayed cord clamping and early cord clamping respectively) are not unique or standard abbreviations in the general medical literature. However, as suggested in the previous response to Referee comments, amongst the readership to which this manuscript is primarily directed, namely the fields of birth transition physiology, obstetrics, neonatology and midwifery, these abbreviations are fairly standard and widely-used. Given that other abbreviations have been largely avoided in the manuscript and that both DCC and ECC have been defined not only at the first point of their usage in the Introduction, but also again in the first paragraph of the Discussion, as well as in the Key Points Summary, the Abstract and

Figure legends, I would contend that a general reader of the manuscript should have little difficulty in recognizing and retaining the meaning of these abbreviations.

Graphs. The majority of figures are vertical bars. Yes, some of the means are very close to zero. But, of course this will be obvious regardless of the method of presentation, and realistically most are not close to the axis. After careful consideration, I submit that superimposed vertical scatterplots showing the actual data would give the reader much more information. This should be a discussion with the Editors.

Many versions of the figures, including scatterplot presentations, were examined after the previous comments of the Reviewing Editor and this Referee. The column graphs submitted in the last revision were considered to be the best overall compromise for presentation of a difficult set of study data that included 18- to 43-fold increases in two key variables (pulmonary blood flow and pulmonary vascular conductance) between the fetal and newborn periods. A major advantage of these column graphs was that they provided a clear distinction between changes in all variables examined in the seven defined phases of the birth transition occurring with 1) early cord clamping followed by a non-asphyxial interval before ventilation, 2) early cord clamping followed by an asphyxial interval before ventilation, or 3) initial ventilation followed by delayed cord clamping. Note that, over these cord clamping strategies, a total of nine different phases needed to be reliably distinguished (including two time-points after early cord clamping and before ventilation, and two time-points after initial ventilation but before delayed cord clamping).

In light of the above most recent comment of the Referee, a vertical scatterplot presentation of the study data has been re-evaluated. Unfortunately, this presentation of the study data has two main drawbacks, which particularly affect the early non-ventilatory phases of the birth transition:

- 1) As mentioned in the previous response, with the size of readily-visible symbols employed in vertical scatter plots, symbols of datapoints that are close to zero or quite small in magnitude (e.g. fetal baseline pulmonary blood flow and pulmonary vascular conductance in Fig. 3, left-to-right ductal flow before cord clamping in the early cord clamping groups or before ventilation in the delayed cord clamp group in Fig. 6) sit over and extend below the x-axis in a clustered aggregation. This is not only unappealing to the eye, but also makes it well-nigh impossible to distinguish individual data-points within this cluster. It is thus very difficult to see how this presentation would provide more information to a reader about these phases (which are a main focus of the manuscript) than the present column format. As pointed out by the Referee, a vertical scatterplot presentation is more informative with respect to the distribution of datapoints with other (much higher) levels of pulmonary blood flow and pulmonary vascular conductance present after birth. However, from the point of view of the main focus of the manuscript, these are less important than the earlier non-ventilatory phases.
- 2) Only a limited number of symbols are available for scatter plots (e.g. open or filled in circles, triangles, squares etc). However, particularly with datapoints that are close to zero or quite small in magnitude, clear delineation between different symbols is almost impossible in the three groups, because for example, open symbols are clustered together in very close proximity and thus appear as though they are filled in. This precludes a clear distinction between effects of different phases of the birth transition in many of the graphs and re-introduces other issues that were a problem with figures in the initial submission, e.g. the need to position phase labels below the x-axis.

For these reasons, re-evaluation of graphical representation of study data has reaffirmed the view that the column graphs submitted in the last revision are the best overall compromise for presentation of a difficult set of data in figures of this manuscript. It is also important to note that, if readers require more detailed information about study data, then these data can be accessed via the Data Availability mechanism.

Dear Associate Professor Smolich,

Re: JP-RP-2025-288749R2 "Non-ventilatory events substantially contribute to rise in pulmonary arterial blood flow with early or delayed cord clamping at birth in preterm lambs" by Joseph John Smolich

We are pleased to tell you that your paper has been accepted for publication in The Journal of Physiology.

Yours sincerely,

Laura Bennet
Senior Editor
The Journal of Physiology

If you would like to receive our 'Research Roundup', a monthly newsletter highlighting the cutting-edge research published in The Physiological Society's family of journals (The Journal of Physiology, Experimental Physiology, Physiological Reports, The Journal of Nutritional Physiology and The Journal of Precision Medicine: Health and Disease), please click this link, fill in your name and email address and select 'Research Roundup':
<https://www.physoc.org/journals-and-media/membernews>

- **TRANSPARENT PEER REVIEW POLICY:** To improve the transparency of its peer review process, The Journal of Physiology publishes online as supporting information the peer review history of all articles accepted for publication. Readers will have access to decision letters, including Editors' comments and referee reports, for each version of the manuscript as well as any author responses to peer review comments. Referees can decide whether or not they wish to be named on the peer review history document.
- You can help your research get the attention it deserves! Check out Wiley's free Promotion Guide for best-practice recommendations for promoting your work at: www.wileyauthors.com/eoo/guide. You can learn more about Wiley Editing Services which offers professional video, design, and writing services to create shareable video abstracts, infographics, conference posters, lay summaries, and research news stories for your research at: www.wileyauthors.com/eoo/promotion.
- **IMPORTANT NOTICE ABOUT OPEN ACCESS:** To assist authors whose funding agencies mandate public access to published research findings sooner than 12 months after publication, The Journal of Physiology allows authors to pay an Open Access (OA) fee to have their papers made freely available immediately on publication.
